# Semantic-KG: Using Knowledge Graphs to Construct Benchmarks for Measuring Semantic Similarity

**Qiyao Wei**[*]
University of Cambridge

**Edward Morrell**
GSK.ai

**Lea Goetz**
GSK.ai

**Mihaela van der Schaar**
University of Cambridge

## Abstract

Evaluating the open-form textual responses generated by Large Language Models (LLMs) typically requires measuring the semantic similarity of the response to a (human generated) reference. However, there is evidence that current semantic similarity methods may capture syntactic or lexical forms over semantic content. While benchmarks exist for semantic equivalence, they often suffer from high generation costs due to reliance on subjective human judgment, limited availability for domain-specific applications, and unclear definitions of equivalence. This paper introduces a novel method for generating benchmarks to evaluate semantic similarity methods for LLM outputs, specifically addressing these limitations. Our approach leverages knowledge graphs (KGs) to generate pairs of natural-language statements that are semantically similar or dissimilar, with dissimilar pairs categorized into one of four sub-types. We generate benchmark datasets in four different domains (general knowledge, biomedicine, finance, biology), and conduct a comparative study of semantic similarity methods including traditional natural language processing scores and LLM-as-a-judge predictions. We observe that the sub-type of semantic variation, as well as the domain of the benchmark impact the performance of semantic similarity methods, with no method being consistently superior. Our results present important implications for the use of LLM-as-a-judge in detecting the semantic content of text. Code is available at `https://github.com/QiyaoWei/semantic-kg` and the dataset is available at `https://huggingface.co/datasets/QiyaoWei/Semantic-KG`.

## 1 Introduction

Large language models (LLMs) are increasingly being adopted across diverse domains [3], where their usage is shifting beyond simple question-answering, towards applications involving language parsing and understanding [10]. For example, LLMs in retrieval-augmented generation (RAG) applications might be required to review and synthesize large quantities of text, containing potentially contradictory findings, to identify passages relevant to a user's query. In addition, LLMs are being used to verify the outputs of other text generation methods, such as the LLM-as-a-judge paradigm, which asks LLMs to simulate ground-truth feedback when the problem is not easily verifiable. It is clear that having robust and reliable LLM evaluation pipelines are crucial in these situations [23] otherwise we run the risk of deploying models that perform suboptimally in real-life applications, potentially leading to incorrect decisions, user dissatisfaction, or even serious consequences in high-stakes applications like biomedicine [34].

When parsing large textual corpora it is important these LLMs detect semantic rather than syntactic content. Two pieces of text might be syntactically similar, in terms of their token-content, but semantically dissimilar, in terms of their overall meaning (see Figure 1). There is a rich suite of semantic similarity methods from the natural language processing (NLP) field, such as token-based methods (e.g. ROUGE, BLEU) and language-embedding models [15, 21].

---

[*]Corresponding author qw281@cam.ac.uk

39th Conference on Neural Information Processing Systems (NeurIPS 2025) Track on Datasets and Benchmarks.

However commonly used metrics such as ROUGE and BLEU have been shown to capture surface similarity while ignoring semantic content and can fail in the presence of small perturbations to text that alter their meaning [24, 9, 18]. As LLMs are increasingly deployed to diverse applications requiring deep textual understanding, it is essential to ensure that they can capture subtle variations in the semantic meaning of text. Additionally, in the LLM-as-a-judge paradigm, it is important for evaluation metrics to reflect the semantic content of responses rather than rewarding superficial similarities. There is therefore a need for high-quality semantic similarity benchmarks, i.e. benchmarks that can test semantic similarity methods across a range of diverse domains.

Several benchmark datasets exist to test a model's ability to detect semantic equivalence between text, such as the STS benchmark, Winograd, and MRPC [2, 12?, 8]. There are also domain-specific benchmarks like BIOSSES for testing semantic equivalence in the biomedical domain [31]. However, these benchmarks are often costly to generate, relying on expensive human judgement [5, 4]. Furthermore, it is often not clear how semantic equivalence is defined, with several of these bench-

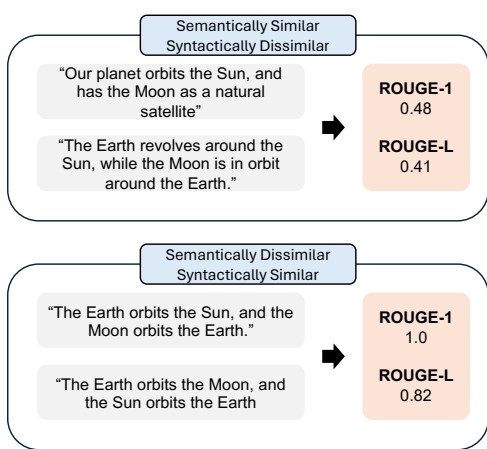

Figure 1: **Difference between semantic and syntactic variations.** Two text samples that are syntactically different but semantically equivalent (top), and syntactically similar but semantically different (bottom). ROUGE-1 and ROUGE-L scores are shown for each statement pair, highlighting the limitations of these methods to detect semantic meaning.

marks relying on subjective human judgement rather than a clear notion of semantic equivalence [40]. Despite efforts to reduce human judgment subjectivity via annotation guidelines and structured scores such as Likert scales [2], human annotations may still be conflicting or unavailable for a specific domain.

**Contribution**  In this paper we introduce a scalable knowledge-graph (KG) based framework for generating semantic similarity benchmark data for any domain. The key advantage of our approach is that we can specify the semantic content of a natural language statement and directly control the semantic similarity/dissimilarity through KG perturbations, allowing us to generate a semantic similarity benchmark without having to rely on human judgment. Our contributions in this paper are threefold: **(C1)** We introduce the Semantic-KG framework that can generate semantic benchmark data containing targeted semantic variations in text. **(C2)** We publish a version of this dataset containing text from four different domains: General-knowledge, Biomedicine, Finance, and Biology. **(C3)** We assess the performance of several semantic-similarity methods including LLM-as-a-judge, and traditional NLP methods on our benchmark, presenting a comparison of how different methods compare at evaluating the semantic similarity of statements from various domains and contexts. The paper is organised as follows: In section 2, we present our semantic similarity benchmark based on KGs. Then, we summarize the works related to benchmarks for measuring semantic similarities. Finally, we benchmark the performance of semantic-similarity methods across the generated datasets.

## 2   The Semantic-KG Framework

Our framework seeks to generate benchmark datasets to evaluate the ability of semantic textual similarity (STS) methods such as LLM-as-a-judge, ROUGE, BLEU etc. to detect variations in the semantic content of text. We use knowledge graphs (KGs) which represent the semantic relationships between entities across diverse domains. A KG represents relationships as a graph, storing the entities as nodes and the relationships as edges. The data can be stored in a triple format consisting of a source-node, relation, and target-node.

Generating our benchmark involves several key stages (see Figure 2). First, we apply perturbations to KGs to subtly alter the semantic relationships between entities, applying one of 4 perturbation-types, corresponding to different types of semantic variation in text. Next we use an LLM to generate textual statements grounded by the original and perturbed KGs. These generated statements are used to form similar and dissimilar statement pairs that can be used as an evaluation benchmark dataset (see Figure

3). To ensure the quality of all LLM generated statements, each statement undergoes a validation step to ensure we can reconstruct the KG using the generated text. Our framework can be applied to any domain for which KGs are available, enabling the scalable generation of semantic-similarity evaluation benchmarks across a wide range of domains.

In this section we detail the four key steps of the proposed method: 1) Subgraph sampling, 2) Subgraph perturbation, 3) Response generation, 4) Response validation.

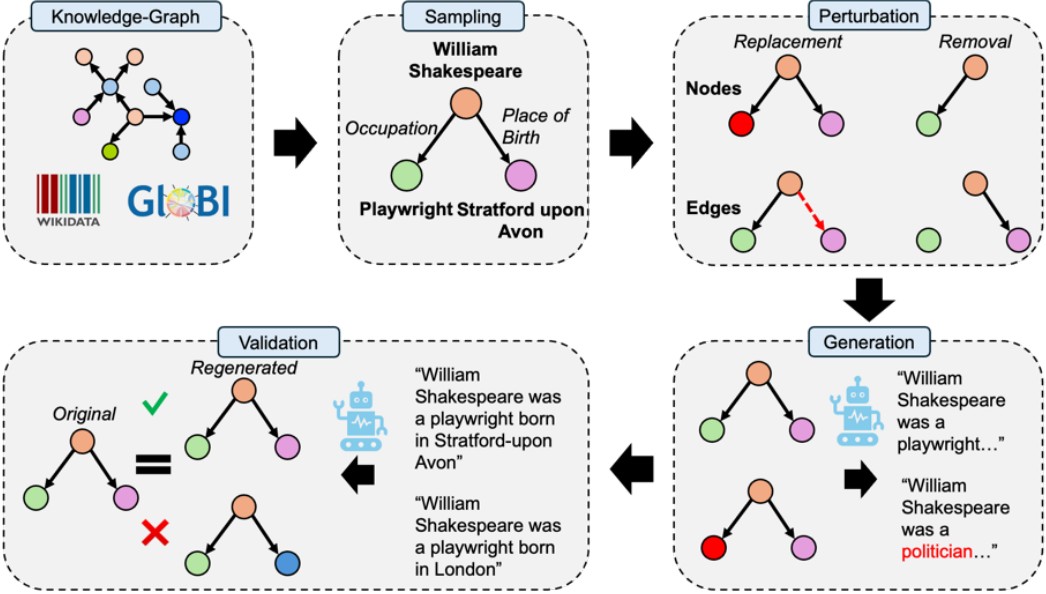

Figure 2: **Overview of the Semantic KG Framework.** Semantic KG consists of 4 stages: 1) Sampling: A subgraph is sampled from a knowledge-graph dataset, 2) Perturbation: The knowledge-graph is perturbed, 3) Generation: Textual statements are generated from the subgraph and perturbed subgraphs, 4) Validation: Statements are validated for correctness using reconstruction accuracy.

## 2.1 Subgraph sampling

In the first stage of generation we sample a subgraph from a KG database. A KG database stores knowledge in a particular domain in the form of nodes describing individual entities, and edges describing relationships between entities.

As a first step, we randomly select a node in the KG as the seed node. Then, we perform subgraph sampling by traversing the node's neighbors in a breadth-first search (BFS) manner, such that between 5 and 20 neighbors are randomly selected for exploration. To ensure diversity of nodes, we also reduce the selection probability of visiting nodes of a given type (e.g. nodes of type "Person") once a node of a that type has been visited. We use BFS rather than depth-first search (DFS) for the same reason, namely to encourage exploration in node diversity when sampling the subgraph. The total number of nodes and edges in the sampled subgraphs are shown in Table 5 and 6.

## 2.2 Subgraph perturbation

We next introduce a set of perturbations into the sampled subgraphs to modify the semantic content of the subgraph. The goal of a perturbation is to alter the semantic content of a subgraph in a targeted manner. The perturbations we apply either remove existing information, or modify information. Perturbations are applied both to entities contained within the subgraph, through perturbations to nodes, and the relationships between entities, by targeting edges. We apply perturbations of the following types:

- Node Removal: A node is randomly removed from the subgraph. All edges between the removed node and its neighbours are also removed. Nodes are only removed if connected nodes in the subgraph have at least 2 neighbours to avoid creating isolated nodes.

- Node Replacement: A node is randomly replaced with another node and all edges to and from the old node are modified to point to and from the new node. To ensure the sampled

node maintains logical consistency with the existing subgraph, the node is only replaced by nodes of same type. For example a node of type "Person" would only be replaced by another person.

- Edge Removal: An edge is randomly removed between 2 nodes. As with node removal, an edge is only removed if both nodes have at least 2 edges to avoid creating an isolated node.

- Edge Replacement: The value of an edge is randomly replaced with a new value. An edge-replacement value is chosen such that it modifies the meaning of the triple. For example, for the triple ("benidipine", "increases effect", "bradycardia"), a valid edge-replacement value for "increases effect" might be "decreases effect". Or for the triple ("Henry VIII", "child", "Elizabeth I"), "parent" or "spouse" would be a valid replacement for "child" as this fundamentally alters the relationship. For each dataset a custom mapping was manually defined, specifying the allowed edge-replacements for a given edge-value. Please see Appendix H for the defined mapping for each dataset.

A random number of perturbations between 1 and 70% of the total number of nodes in the graph was applied to any given subgraph, to ensure variability in the semantic similarity between a subgraph and perturbed subgraph pair.

For the benchmark presented in this paper our choice of perturbation was arbitrary, aiming to address common forms of semantic variation such as missing or conflicting information. However, the flexibility of our framework allows for the incorporation of new perturbations tailored to specific applications, such as the introduction of noise or irrelevant information, in future benchmarks.

## 2.3 Response generation

In this stage we convert the subgraph and perturbed subgraphs into natural-language statements using an LLM. First, each subgraph is converted into a triple format, consisting of "source-node", "relation", and "target-node" tuples. The number of triples per subgraph is equal to the number of edges (see Tables 5 and 6). An LLM is then instructed to generate a natural-language statement using all provided triples. Few-shot examples are used to encourage the LLM to express logical relationships that may not be explicitly represented by individual triples. For example given the triples: ("Norway", "member of", "Organisation for the Prohibition of Chemical Weapons") and ("Colombia", "member of", "Organisation for the Prohibition of Chemical Weapons"), the LLM might choose to represent this as a single statement: "Both Norway and Colombia are members of the Organisation for the Prohibition of Chemical Weapons". Please see Appendix I.2 for the full prompt used for each dataset.

We refer to the generated statements for the original subgraphs as "original statements", and the generated statements for the perturbed subgraph as "perturbed statements". For each subgraph, we generate at least two different original statements to form semantically similar pairs and one perturbed statement containing one of the four previously mentioned perturbations to form a semantically dissimilar pair (using one of the original statements as the other statement in the dissimilar pair).

For details on the models and parameters used for generation please see Appendix C.1.

## 2.4 Response validation

Finally, we perform validation on each statement to ensure its quality. In this stage we use an LLM to reconstruct a subgraph using a generated statement. This reconstructed KG is compared for equivalence to the original subgraph to score it for correctness. Response validation occurs in two stages: entity-extraction and KG-extraction. In the first stage an LLM is provided with a generated statement and a list of entity-types (e.g. "Person", "Place", etc.) and instructed to extract all entities from the generated statement. The list of entity-types is generated on a per-dataset basis based on all types within the full KG dataset. The entity-extraction prompt uses a one-shot example. In the second stage an LLM is provided with the response, the entities extracted in the entity-extraction stage, and a list of valid edge-types. As with entity-extraction, the list of valid edges is generated on a per-dataset basis based on the full KG dataset. The LLM is then instructed to generate triples describing the relationships between all the provided entities using the generated statement. The KG-extraction prompt uses three-shot examples. For full details of the prompts, models and generation parameters used for entity-extraction and KG-extraction, see Appendix C.1 and I.3.

When comparing the reconstructed triples to the original triples, post-processing is applied to both sets of triples. In this step all entities are made lower-case, stop-words and whitespace are removed,

and each entity is lemmatized. This ensures the comparison does not penalize trivial differences between entities, such as "The United Kingdom" and "United Kingdom". A generated response is only considered valid if the reconstructed triples exactly match the original triples. This ensures that all generated statements accurately reflect the KG structure of the original subgraph.

# 3 Dataset Overview

Table 1: **Summary Statistics for Semantic-KG Dataset**. See Appendix D for extended statistics

| Dataset Name | Perturbation Type | Number of Statements | Avg Word Count |
|---|---|---|---|
| Codex | Edge Deletion | 114 | 158.94 |
| | Edge Replacement | 82 | 154.79 |
| | Node Removal | 134 | 141.46 |
| | Node Replacement | 171 | 143.96 |
| FinDKG | Edge Deletion | 75 | 120.45 |
| | Edge Replacement | 127 | 112.35 |
| | Node Removal | 159 | 100.27 |
| | Node Replacement | 135 | 112.90 |
| Globi | Edge Deletion | 167 | 205.99 |
| | Edge Replacement | 145 | 199.19 |
| | Node Removal | 200 | 189.70 |
| | Node Replacement | 148 | 205.87 |
| Oregano | Edge Deletion | 218 | 118.03 |
| | Edge Replacement | 478 | 86.91 |
| | Node Removal | 174 | 105.25 |
| | Node Replacement | 162 | 123.65 |

In this section, we describe the 4 KG datasets used to generate the benchmark dataset we publish with this manuscript. The flexibility of this framework will allow this benchmark to be expanded to any KG dataset from any domain. To generate a semantic similarity benchmark for a new domain, a KG must contain triples consisting of the name and type of a source and target node, and a description or name of the edge between the source and target node. Summary statistics for our benchmark are shown in Table 1.

- **Codex**[26] A general-knowledge KG based on WikiData [37] containing 77,951 entities and 69 relations based on Wikipedia data.

- **Oregano**[1] A biomedical KG containing data on relationships between drug compounds and biological entities. The dataset contains 11 types of node and 18 types of edge.

- **FinDKG**[14] A financial KG containing 13,645 entities and 15 relations representing global economic and market trends.

- **Globi**[22] A biology KG containing global biotic interaction data such as predator-prey relationships and pollinator-plant relationships.

# 4 Dataset Validation

To validate the quality of our generated dataset we performed analyses to evaluate the generated statements for correctness and linguistic naturalness.

**Correctness:** To validate the robustness of the response-validation step, we performed a manual spot-check experiment where a human annotator manually scored 100 generated statements (50 from original subgraphs; 50 from perturbed subgraphs) for correctness (see Table 2). A statement was evaluated as correct if it accurately and completely reflected the semantic content of its source subgraph without introducing extraneous information. 99% of evaluated statements were evaluated as correct.

**Naturalness:** We next evaluated linguistic naturalness to ensure the generated data resembled real-world text. First, we analysed generated responses using a suite of standard NLP metrics to assess the readability, lexical characteristics, and syntactic complexity of the generated data (see Appendix E). The statements matched academic paper readability (Flesch 10-30 range), and exhibited high lexical diversity, typical of specialized academic writing. Additionally, statements were similar in syntactic complexity to academic literature, though with shorter sentences and slightly lower noun ratios. We additionally performed a manual spot-check for naturalness (see Table 2). Two human annotators independently scored the same 100 statements from the correctness check, rating each for linguistic fluency and "naturalness". A statement was considered natural if both annotators agreed. In this evaluation, 75% of statements were rated as natural.

Table 2: **Summary of Manual Spot-Check Validation Results for the Generated Dataset.**

| Metric | Sample Size | Num. Annotators | Result (%) |
|---|---|---|---|
| Correctness | 100 | 1 | 99 |
| Naturalness | 100 | 2 | 75 |

## 5   Related Works

In this section we summarize the related works focusing on four areas (1) Automated pipelines to generate LLM evaluation data (2) LLM-as-a-judge evaluation data (3) Semantic similarity benchmarks (4) Using KGs for LLM validation.

**Automated pipelines to generate LLM evaluation data**   Several works propose automated pipelines to generate LLM evaluation data. For instance, [7] use word-order perturbations to create natural-language inference data that introduce contradictions into textual statements. [27] uses sentence templates to introduce perturbations into text that impact on properties of the text such as introducing contradictions or irrelevant information. Additionally, [32] uses WikiData [37] to introduce different types of knowledge conflicts into text to assess the impact on LLM behaviour. Despite the success of these works they largely consist of simple English-language or general-knowledge statements that may not be applicable to more complex domains such as biomedicine or finance.

**LLM-as-a-judge evaluation data**   Several benchmarks exist with applications for LLM-as-a-judge evaluations. For instance, MTBench [41] establishes a platform to evaluate LLMs based on open interactions with users and user ratings. These benchmarks often rely on agreement with subjective human preferences. As such, it is unclear whether these benchmarks specifically measure a model's capability to capture semantic content rather than any other human rating preference. [33] proposes an LLM-as-a-judge benchmark based on factual accuracy that can also be applied to new datasets. This approach is limited to datasets with a ground-truth answer and restricted outputs, such as question-answering. Additionally, these benchmarks only evaluate a judge on its capability to distinguish correct answers from incorrect ones, but not a judge's ability to detect more subtle variations in semantic content.

**Semantic similarity benchmarks**   Within the semantic textual similarity (STS) field there are several widely used benchmarks. SemEval [2] aggregates different benchmarks for evaluating various capabilities of STS methods, such as sentiment analysis, toxicity capture, and temporal relations. [12] proposes the Winograd scheme challenge, consisting of pairs of sentences with a referential ambiguity in the two sentences. Commonly used STS benchmarks typically contain human-annotated sentence pairs, and primarily focus on general domains [17, 19]. There are also a small number of domain-specific STS benchmarks. BIOSSES [31] contains sentences annotated for similarity from the biomedical literature, MedSTS [38] consists of expert-annotated sentence-pairs for clinical applications, and [? ] contains questions related to COVID-19, annotated for similarity. To our knowledge, there are no widely used STS benchmarks specific to domains outside of biomedicine. Additionally, most of these benchmarks rely on human annotation, making them costly to generate, and for some complex domains, finding expert annotators may not be possible.

**Using Knowledge Graphs for LLM validation**   KGs have been used as components in LLM validation systems (e.g. RAG), but there is limited work using KGs to explicitly construct benchmark datasets [13]. [16] also use a KG to construct evaluation data, using KGs to generate simple factual statements for accuracy evaluation in LLMs. Unlike our work they do not focus on constructing benchmark data for assessing semantic similarity, additionally the focus of this work is on factual accuracy and correctness rather than subtle semantic changes.

# 6 Experiments

## 6.1 Task Setup

The generated benchmark dataset consists of semantically similar pairs of statements generated from the same subgraph, labeled 1. These form half of the samples in the dataset. The other half consist of semantically dissimilar pairs of samples generated from a subgraph and perturbed subgraph respectively, labeled 0 (see Figure 3). The dissimilar pairs are further categorized into the 4 perturbation sub-types (section 2.2) to allow the results to be stratified according to semantic variation type. In the evaluation task the semantic-similarity model or method is tasked with predicting the label of the natural-language statement pairs.

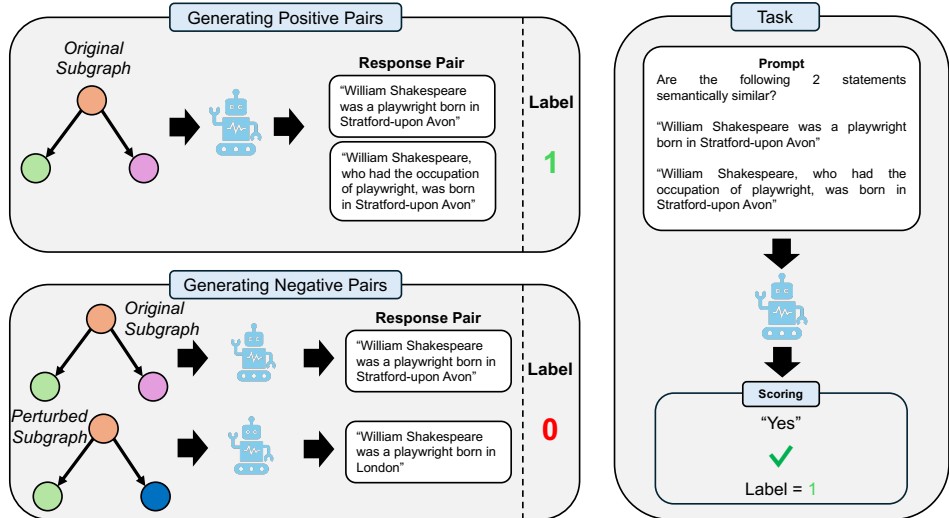

Figure 3: **Overview of the Semantic KG Task.** Positive response pairs *top left*, are generated by sampling 2 responses from the same subgraph. Negative response pairs *bottom left* are generated by sampling a response from the original subgraph and a perturbed subgraph. The model is tasked with predicting the label of the response pairs.

In the Results Section 6.2, we apply this task to 3 types of semantic-similarity method: 1) LLM-as-a-judge, 2) Embedding-models and 3) NLP methods (e.g. ROUGE and BLEU). For LLMs, both statements are formatted into a prompt, and the LLM is asked whether the two statements are semantically similar. For full details of the prompt see Appendix I.1. For embedding models, we embed the two statements and compute the cosine similarity, and for NLP methods we simply compute the metric for the statement pair. For methods that output a continuous score, the score must be converted to a binary label using a threshold. To compute this threshold we split the data into validation and test data and find the threshold that maximizes the F1-score using the validation data. The test data is then used to report the final results. For LLMs we only report results for the test data.

## 6.2 Results

Figures 4 and 5 show the results for different semantic-similarity methods on our benchmark, stratified by perturbation-type and dataset respectively.

**Stratification by perturbation-type** revealed disparities in performance across all semantic similarity methods, depending on the type of perturbation applied with statistical analysis revealing a significant effect of perturbation type for both node-removal ($\beta = 0.124$, $p = 0.039$) and node-replacement ($\beta = 0.155$, $p = 0.01$). Many methods appear to under-perform when distinguishing between statements that differ as a result of edge perturbations compared to node perturbations. Interestingly, the relative superiority of LLMs compared to classic NLP methods appears to depend on the perturbation-type. When perturbing edges, LLMs appear to match or out-perform traditional methods, especially state-of-the-art models, with statistical analyses revealing a significant interaction effect between perturbation-type and method for GPT-4o (GPT4o x edge-replacement: $\beta = 0.184$, $p = 0.031$). However for node perturbations, such as node-deletions, the traditional methods actually out-perform the majority of LLMs with statistical analyses revealing a significant interaction effect

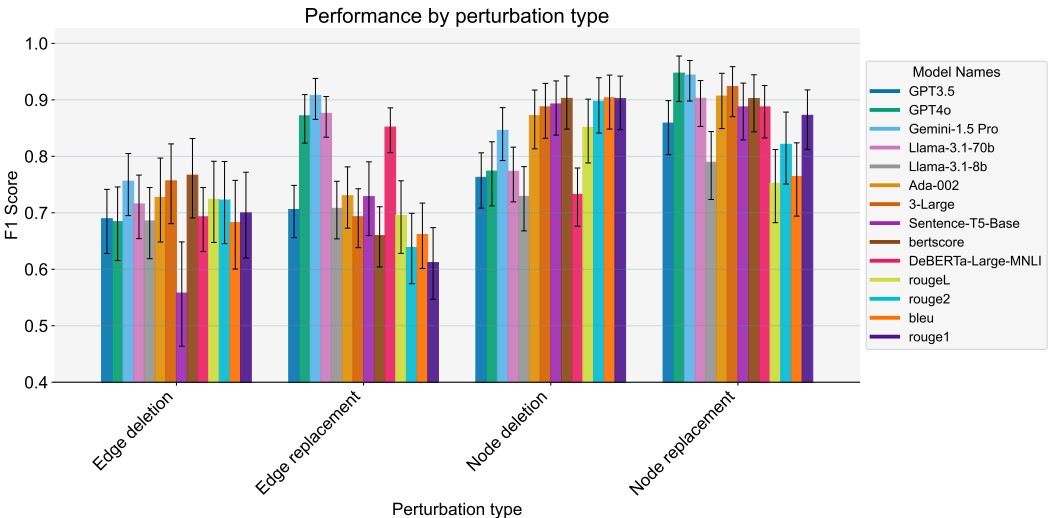

Figure 4: **Semantic Performance by Perturbation-Type.** Performance (F1 Score) of different semantic similarity models stratified by perturbation-type. Error-bars display Clopper-Pearson 95% confidence intervals.

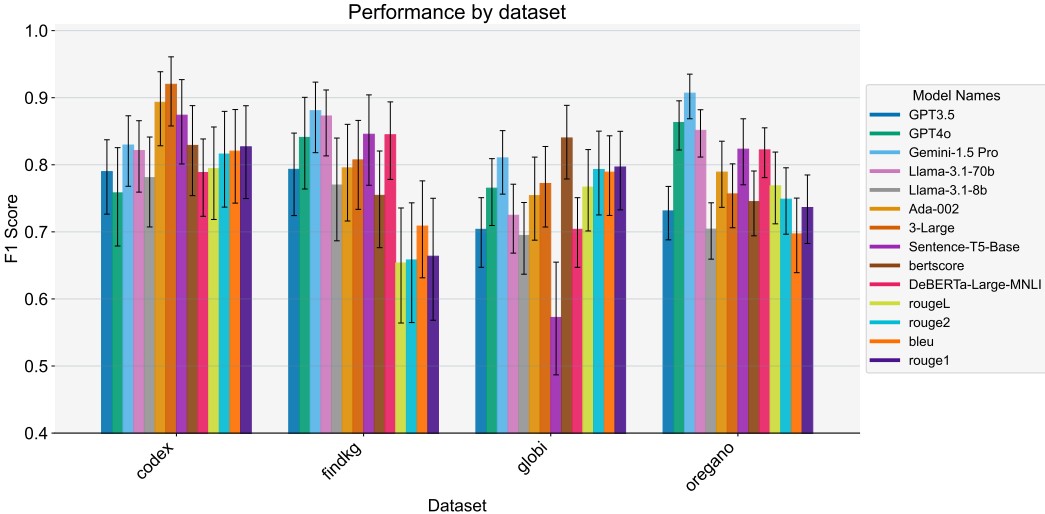

Figure 5: **Semantic Performance by Dataset.** Performance (F1 Score) of different semantic similarity models stratified by dataset. Error-bars display Clopper-Pearson 95% confidence intervals.

between perturbation type and model for Sentence-T5-Base (Sentence-T5-Base x node-removal: $\beta = 0.260$, $p = 0.002$; Sentence-T5-Base x node-replacement: $\beta = 0.216$, $p = 0.011$). See Table 14 for full statistical analysis results. This has important implications for real-world applications of semantic-similarity methods. LLMs might be better suited for applications that require detecting variations in the relationships between entities in a statement, for example detecting contradictions in text. However traditional methods likely suffice for settings that simply involve detecting whether statements encompass the same entities.

**Stratification by dataset** also revealed disparities in performance, though these were less pronounced for LLMs than for perturbation-type disparities, and significant effects were only observed for one interaction (see Table 14 for full results). All methods appeared to show strong performance on the general-knowledge dataset (Codex), with slightly lower performance on domain-specific datasets such as Globi and FinDKG, though a marked performance drop was observed for Sentence-T5-Base (Sentence-T5-Base x Globi: $\beta = -0.413$, $p = 0.002$. Performance disparities were higher among traditional NLP methods with Bertscore outperforming all methods including LLMs on Globi, yet reductions in performance observed for the finance dataset (FinDKG) for methods such as ROUGE1 and ROUGEL, though this effect was not observed to be statistically significant (Rouge-1 x FindKG: $\beta = -0.236$, $p = 0.081$). A closer inspection of results, (see Appendix F.2) reveals that this performance drop was largely driven by poor performance for edge-replacement perturbations, further highlighting the insufficiency of these methods to detect modifications in textual relationships. Additionally, the relative benefit of LLMs appeared to be domain-dependent with some LLMs under-performing for certain domains such as Globi and Oregano, whereas other LLMs such as Gemini-1.5 Pro, showed superior performance on domain-specific datasets like Oregano. These results highlight that semantic-similarity performance in one domain may not necessarily translate to performance in another. As more datasets are adopted into this framework, this will further elucidate the strengths and weaknesses of different methods within domain-specific applications.

By applying our framework to four diverse domains and evaluating a range of semantic similarity methods, we demonstrate that both traditional metrics and modern LLMs can display notable weaknesses — particularly for perturbations such as relationship changes. Our findings emphasize the importance of domain-specific and semantic-aware evaluation, showing that method performance can vary widely depending on the nature of the semantic variation and the target domain.

# 7 Discussion

In this work, we introduced the Semantic-KG framework, a scalable and domain-agnostic method for generating high-quality semantic similarity benchmarks using knowledge graph (KG) perturbations. Our framework enables fine-grained control over semantic variations and supports benchmark creation across any domain with an available KG, without relying on human annotation which is both expensive and subjective. Furthermore, our benchmark enables straightforward comparison of STS methods, addressing the important challenge of automatic and scalable STS dataset generation. In our experiments, we show that performance across all STS methods (embedding models, LLM-as-a-judge, etc) vary depending on the type of KG perturbation and the domain of the KG, and LLMs are not by default the best choice for STS tasks.

Using our novel Semantic-KG framework, we benchmarked several semantic similarity methods, revealing nuances in their performance with important implications for real-world applications of LLMs. The strong performance of NLP methods on node-perturbations suggests they may be sufficient in contexts that simply rely on detecting the presence or absence of entities in text. For example, in a RAG system designed to return documents relevant to a user's query, it may be sufficient for a system to detect whether a document contains information on "Quantum Entanglement" vs. "Quantum Computing". However, if RAG systems are involved in synthesising that information, particularly in high-stakes domains like medicine, then the inability of these methods to detect edge-perturbations may pose risks. For example, an LLM might need to distinguish between a paper stating, "Drug A treats disease X" and another claiming "Drug A causes side-effect similar to disease X", and failure to do so may present a risk to patients.

Furthermore, domain-specific performance variations may present risks in certain applications. An investment firm relying on LLMs to summarize financial reports might use traditional metrics like ROUGE, or LLM-as-a-judge, to validate the relevance of LLM-generated summaries, by comparing these to source documents. The marked decrease in performance of ROUGE on FinDKG, could

lead to an NLP-based verifier failing to detect incorrect information in a report, particularly if that information relates to relationships between entities. Additionally LLMs can underperform in domains like biology (Globi) compared to their general knowledge performance emphasizing that deploying an LLM in a high-stakes field like biomedicine requires rigorous, domain-specific validation.

Our framework and our presented results demonstrate that validating the outputs of LLMs in real-world settings necessitates a granular understanding of semantic content. By introducing a framework that can precisely target different types of semantic variations across diverse domains, our approach facilitates a more informed selection and fine-tuning of LLMs and semantic-similarity tools. This is vital for ensuring that these powerful models perform reliably and safely, preventing incorrect decisions and building user trust, especially as they become increasingly integrated into critical applications. Future work will focus on expanding the diversity of KG datasets and perturbation types to further refine our understanding and evaluation of LLM semantic understanding.

## 8 Limitations & Future Work

This paper presents a novel benchmark for semantic similarity evaluation, though it is not without its limitations.

**(1) Dependency on Knowledge Graph Availability and Quality:** The availability and quality of the generated benchmarks depends on the availability of high-quality KGs. For some domains, KG datasets may not be available, or they may be of low quality containing inaccurate or incomplete data, which will impact the quality of the generated textual pairs. **(2) Scope of Semantic Variations:** Our current benchmark applies perturbations designed to create clear semantic distinctions. However real-world semantic variations are likely complex and not well captured by simple node or edge deletions and replacements, such as variations in intent or implicature, changes in tone, or context-dependent changes in meaning. Our framework is flexible to many perturbation types and future work might seek to apply more complex, realistic perturbations to generate benchmark data. Additionally, we might explore methods to control the semantic closeness of candidate replacement nodes using techniques such as embedding models, to understand how this impacts on the performance of different methods. However it is likely that some real-world semantic variations will not be well captured by structural knowledge-graph perturbations. Nonetheless our benchmark identified key limitations of different semantic-similarity methods highlighting that even simple semantic perturbations still have utility for evaluating these methods. **(3) Statement Validation:** The validation step, relying on KG reconstruction accuracy, primarily ensures the generated text is grounded in and reflects the subgraph. It may not fully capture other aspects of textual quality, such as fluency, naturalness, or the absence of unintended connotations, which could subtly influence how methods perceive similarity. **(4) Limitations of Knowledge-Graphs for encoding semantic knowledge:** One key limitation of using knowledge-graphs to encode semantic knowledge is that they inherently operate under a closed-world assumption [25] whereby missing information is assumed false, however this may not capture the incompleteness or evolving nature of the knowledge encoded in datasets generated using our framework. In future work, a more nuanced evaluation might explore how performance of different semantic-similarity methods varies under an open-world assumption. **(5) Limitations on reliance on LLMs for generation:** Our framework relies on LLMs to generate benchmark data. Though all generated samples are validated and grounded in a knowledge-graph, LLM generated textual data may still differ in distribution from real-world text, potentially introducing subtle biases. This framework is not intended to replace real human-labelled data, though still offers a valuable tool for identifying weaknesses in semantic-similarity methods, where data is scarce or human labelling too costly. **(6) Simple task setup:** Our current benchmark only evaluates semantic-similarity methods in the simple binary setting, however graded annotation schemes inspired by human-labeled datasets [2] might be incorporated into future versions of our dataset, using metrics such as perturbation-count or graph similarity measures [28]. **(7) Limitations of subgraph-size:** The success of the generation and validation pipeline in our framework declines at higher subgraph sizes (>12 triples; see Figure 7), which limits the size of generated statements. As LLMs improve we hope to build on our framework to encompass larger subgraphs, which will allow for the generation of more complex semantic-similarity benchmarks.

**Acknowledgements** We thank Jessica Schrouff for her helpful input on the manuscript. We are also grateful for the valuable feedback from all of the anonymous OpenReview reviewers, and from our colleagues in the van der Schaar lab and at GSK, for their input, comments, and suggestions. QW would like to acknowledge and thank GSK for its support via the PhD scholarship program.

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

# Appendix Contents

- **1. Ethical Disclosure**
- **2. Broader Impact Statement**
- **3. Experimental Details**
- **4. Extended Dataset Statistics**
- **5. Extended Dataset Validation**
- **6. Extended Experimental Results**
- **7. Dataset Licenses**
- **8. Edge-Replacement Mappings**
- **9. Prompt Templates**

# A  Ethical Disclosure

(1) Fairness/Bias Risks: This work used data derived from open-source datasets that may contain biases. Indeed, several works report that Wikidata, upon which Codex is based, exhibits over-representation of certain genders and ethnicities [29, 6], while our analysis into the representation of geographical areas in FindKG (Figure 6), reveals a bias towards higher income countries. Semantic-similarity evaluations conducted using our benchmark, might mask biases in evaluated models or methods. We recommend that evaluations conducted using this benchmark are complemented with dedicated fairness and bias assessments. (2) Data-Quality: All generated statements within our framework undergo validation to ensure their faithfulness to the underlying knowledge-graphs these statements are derived from. Nonetheless, the underlying datasets used within this work may contain data-quality issues [30] such as errors or factual inconsistencies. Evaluations using our benchmark should focus on semantic-similarity performance and not factual accuracy, for which dedicated benchmarks already exist [11]. (3) Environmental Impact: Generation of our benchmark dataset required three LLM calls per generated statement: one for generation and two for each stage of response validation, totalling approximately 24,000 LLM calls per dataset. Expanding this framework to new datasets could lead to substantial carbon emissions. The high number of LLM calls reflects the currently low success rate of response validation. We anticipate that improvements to generation and validation will reduce the required number of calls, minimising the environmental footprint of our framework. (4) Misuses: We emphasize that this benchmark is designed for evaluating semantic similarity and should not be used to train new models where doing so risks reinforcing biases and factual inaccuracies present in the source data.

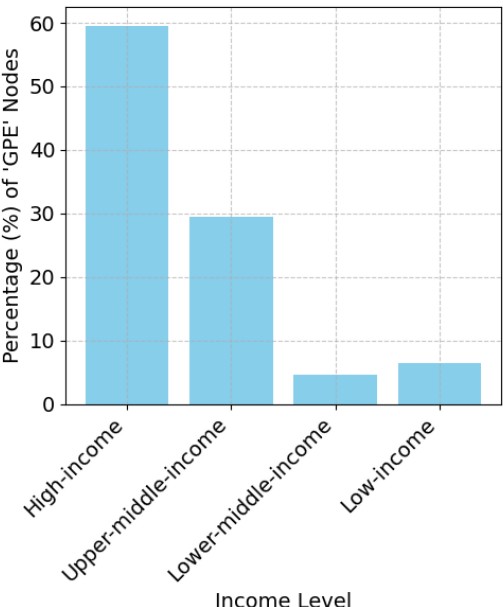

Figure 6: **Distribution of Country Nodes by Income-Level in FindKG** Percentage of Total Geopolitical Entity Nodes ('GPE') in FindKG belonging to different income-levels according to the World-Bank income groups classification [39]

# B  Broader Impact Statement

**Impact Statement**   This paper introduces a framework whose goal is to advance the field of machine-learning. As LLMs are deployed to high-stakes domains it is essential we can validate their outputs, particularly in domains where incorrect outputs may pose risks. Many methods exist for validating language-model outputs, but it is not clear whether these methods can detect nuanced semantic variations. Semantic-textual similarity benchmarks exist for this purpose but are often limited in scope and generating new benchmarks for novel domains may be costly. We introduced this

framework to aid researchers and developers to better explore the strengths and limitations of methods such as LLM-as-a-judge, identifying potential failure modes. Our existing benchmark may provide some initial insights that could be significant for LLM applications in domains such as biomedicine and finance, but we also hope researchers can expand this benchmark to new domains using our framework. Finally, our benchmark is not without its risks and performance on our benchmark is no guarantee that this performance will translate into real-world performance. Researchers should use domain-knowledge and incorporate risk-mitigation strategies into any LLM application, in addition to using benchmarks such as this one.

# C   Experimental Details

## C.1   Model Generation Parameters

All models used in this manuscript, for generation, validation, and evaluation are shown in Table 3 For *response generation* and *response validation* (both entity and KG extraction) we used GPT-4o with a temperature of 1.0.

For model evaluation all models were evaluated with a temperature of 0.0.

| Model Name | Provider | Version |
|---|---|---|
| GPT3.5 | Azure OpenAI | gpt-35-turbo (0125) |
| GPT-4o [20] | Azure OpenAI | gpt-4o (2024-08-06) |
| Gemini-1.5-Pro [35] | Google (VertexAI) | gemini-1.5-pro-001 |
| Llama-70b [36] | Azure (Custom Endpoint) | Meta-Llama-3.1-70B-Instruct |
| Llama-8b [36] | Azure (Custom Endpoint) | Meta-Llama-3.1-8B-Instruct |

Table 3: LLM Model Details

| Model Name | Provider | Version |
|---|---|---|
| Text-embedding-3-large | Azure OpenAI | text-embedding-3-large |
| Text-embedding-ada-002 | Azure OpenAI | text-embedding-ada-002 |

Table 4: Embedding Model Details

## C.2   Confidence-Intervals

To compute confidence-intervals for the F1 score, we first used the Clopper-Pearson method to compute upper and lower bounds for precision and recall respectively. Then we computed the F1-score using every combination of upper and lower bound for precision and recall, and the minimum and maximum F1-score were taken as the upper and lower bounds for the final confidence-interval.

# D    Extended Dataset Statistics

Table 5: **Extended Summary Statistics (1) for Semantic-KG Dataset**

| dataset name | perturbation type | Avg. Subgraph Nodes | Avg. Subgraph Edges |
|---|---|---|---|
| codex | edge deletion | 13.11 | 12.11 |
|  | edge replacement | 12.52 | 11.52 |
|  | node removal | 13.18 | 12.18 |
|  | node replacement | 11.56 | 10.56 |
| findkg | edge deletion | 8.03 | 7.03 |
|  | edge replacement | 6.40 | 5.40 |
|  | node removal | 6.47 | 5.47 |
|  | node replacement | 6.18 | 5.18 |
| globi | edge deletion | 14.18 | 13.18 |
|  | edge replacement | 13.96 | 12.96 |
|  | node removal | 14.07 | 13.07 |
|  | node replacement | 13.98 | 12.98 |
| oregano | edge deletion | 13.36 | 12.36 |
|  | edge replacement | 8.95 | 7.95 |
|  | node removal | 13.05 | 12.05 |
|  | node replacement | 13.25 | 12.25 |

Table 6: **Extended Summary Statistics (2) for Semantic-KG Dataset**

| dataset name | perturbation type | Avg. Perturbed Subgraph Nodes | Avg. Perturbed Subgraph Edges |
|---|---|---|---|
| codex | edge deletion | 13.11 | 9.50 |
|  | edge replacement | 12.52 | 11.52 |
|  | node removal | 5.94 | 4.57 |
|  | node replacement | 11.54 | 10.56 |
| findkg | edge deletion | 8.03 | 5.52 |
|  | edge replacement | 6.40 | 5.40 |
|  | node removal | 3.23 | 2.19 |
|  | node replacement | 6.17 | 5.18 |
| globi | edge deletion | 14.18 | 10.41 |
|  | edge replacement | 13.96 | 12.96 |
|  | node removal | 6.25 | 4.75 |
|  | node replacement | 13.97 | 12.98 |
| oregano | edge deletion | 13.36 | 9.95 |
|  | edge replacement | 8.95 | 7.95 |
|  | node removal | 5.68 | 4.33 |
|  | node replacement | 13.25 | 12.25 |

# E    Extended Dataset Validation

Table 7: **Readability:** Summary of readability for Semantic-KG compared against standard benchmarks for academic and technical documents

| Metric | Semantic-KG | Academic Papers | Technical Docs | Assessment |
|---|---|---|---|---|
| Flesch Reading Ease | 16.2 ± 14.3 | 10-30 | 20-40 | Academic |
| Grade Level | 15.0 ± 2.2 | 14-18 | 12-16 | Academic |
| Gunning Fog | 19.4 ± 3.0 | 15-20 | 12-18 | Academic |

Table 8: **Lexical Characteristics:** Summary of lexical characteristics of Semantic-KG compared against standard benchmarks for academic and technical documents

| Metric | Semantic-KG | Academic Papers | Technical Docs | Assessment |
|---|---|---|---|---|
| Type-Token Ratio | 0.673 ± 0.110 | 0.50-0.70 | 0.45-0.65 | Academic |
| Lexical Density | 0.693 ± 0.058 | 0.65-0.75 | 0.60-0.70 | Academic |
| Unique Words | 48 ± 20 | 40-80* | 30-60 | Academic |

Table 9: **Syntactic Complexity:** Summary of syntactic complexity of Semantic-KG compared against standard benchmarks for academic and technical documents

| Metric | Semantic-KG | Academic Papers | Technical Docs | Assessment |
|---|---|---|---|---|
| Avg Sentence Length | 16.4 ± 3.9 words | 20-30 | 15-25 | Below academic |
| Noun Ratio | 0.232 ± 0.082 | 0.25-0.35 | 0.20-0.30 | Below academic |
| Verb Ratio | 0.095 ± 0.026 | 0.08-0.12 | 0.10-0.15 | Academic |
| Noun-Verb Ratio | 2.44 | 2.5-4.0 | 2.0-3.0 | Technical |
| Content Word Density | 38.0% | 35-45% | 40-50% | Academic |

## E.1 Reconstruction Success Rate

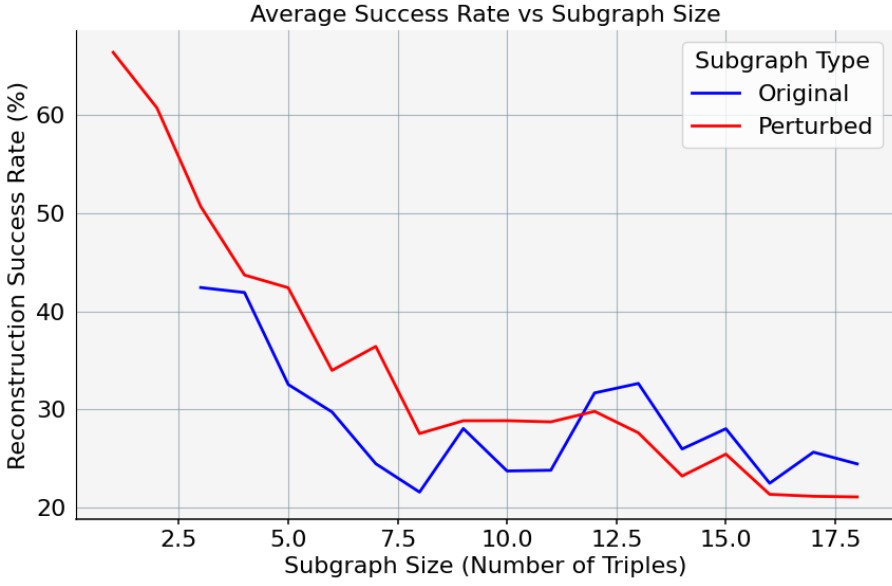

Figure 7: **Reconstruction Success by Subgraph Size.** Average reconstruction accuracy, defined as the percentage of knowledge-graphs successfully reconstructed from generated responses, as a function of subgraph size. *Note:* Only successfully reconstructed subgraphs were used in the final version of Semantic-KG

# F Extended Experimental Results

## F.1 Codex Experimental Results

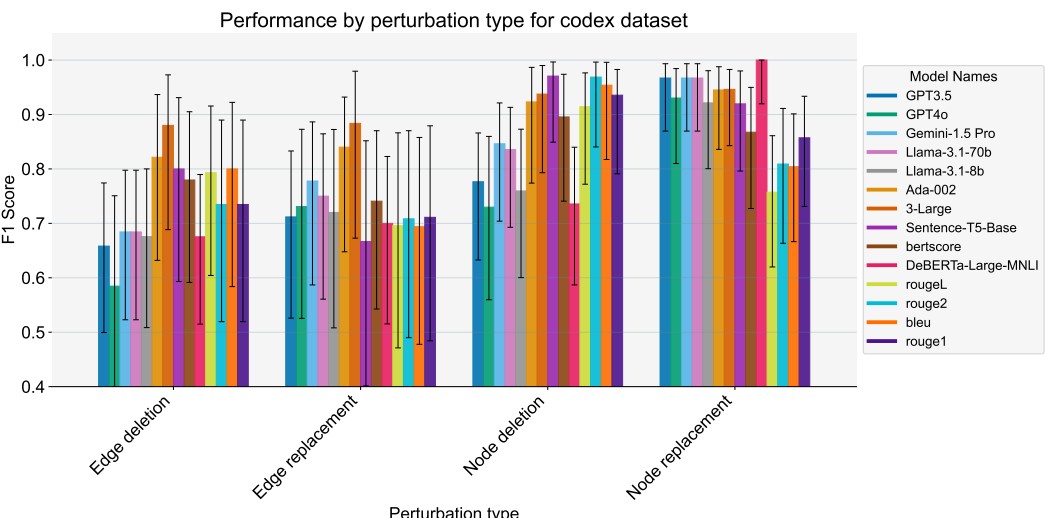

Figure 8: **Semantic Performance by Perturbation-Type for Codex Dataset.** Performance (F1 Score) of different semantic similarity models on the Codex dataset stratified by perturbation-type.

Table 10: Performance of different models on CODEX dataset across perturbation types. Values shown as F1 score ± standard error.

|  | Edge deletion | Edge replacement | Node deletion | Node replacement |
|---|---|---|---|---|
| 3-Large | $0.880 \pm 0.073$ | $0.884 \pm 0.078$ | $0.937 \pm 0.050$ | $0.946 \pm 0.036$ |
| Ada-002 | $0.821 \pm 0.078$ | $0.840 \pm 0.072$ | $0.923 \pm 0.054$ | $0.945 \pm 0.039$ |
| DeBERTa-Large-MNLI | $0.675 \pm 0.070$ | $0.700 \pm 0.078$ | $0.736 \pm 0.065$ | $1.000 \pm 0.021$ |
| GPT3.5 | $0.658 \pm 0.070$ | $0.712 \pm 0.078$ | $0.776 \pm 0.060$ | $0.967 \pm 0.032$ |
| GPT4o | $0.585 \pm 0.089$ | $0.731 \pm 0.089$ | $0.730 \pm 0.077$ | $0.930 \pm 0.044$ |
| Gemini-1.5 Pro | $0.684 \pm 0.070$ | $0.778 \pm 0.076$ | $0.846 \pm 0.055$ | $0.967 \pm 0.032$ |
| Llama-3.1-70b | $0.684 \pm 0.070$ | $0.750 \pm 0.077$ | $0.835 \pm 0.056$ | $0.967 \pm 0.032$ |
| Llama-3.1-8b | $0.676 \pm 0.074$ | $0.720 \pm 0.093$ | $0.759 \pm 0.070$ | $0.921 \pm 0.046$ |
| Sentence-T5-Base | $0.800 \pm 0.086$ | $0.667 \pm 0.115$ | $0.971 \pm 0.038$ | $0.920 \pm 0.047$ |
| bertscore | $0.780 \pm 0.080$ | $0.741 \pm 0.084$ | $0.896 \pm 0.059$ | $0.867 \pm 0.057$ |
| bleu | $0.800 \pm 0.086$ | $0.694 \pm 0.097$ | $0.954 \pm 0.046$ | $0.804 \pm 0.060$ |
| rouge1 | $0.735 \pm 0.094$ | $0.711 \pm 0.101$ | $0.935 \pm 0.049$ | $0.857 \pm 0.052$ |
| rouge2 | $0.735 \pm 0.094$ | $0.708 \pm 0.097$ | $0.969 \pm 0.040$ | $0.809 \pm 0.063$ |
| rougeL | $0.793 \pm 0.079$ | $0.696 \pm 0.101$ | $0.914 \pm 0.052$ | $0.757 \pm 0.062$ |

## F.2 FinDKG Experimental Results

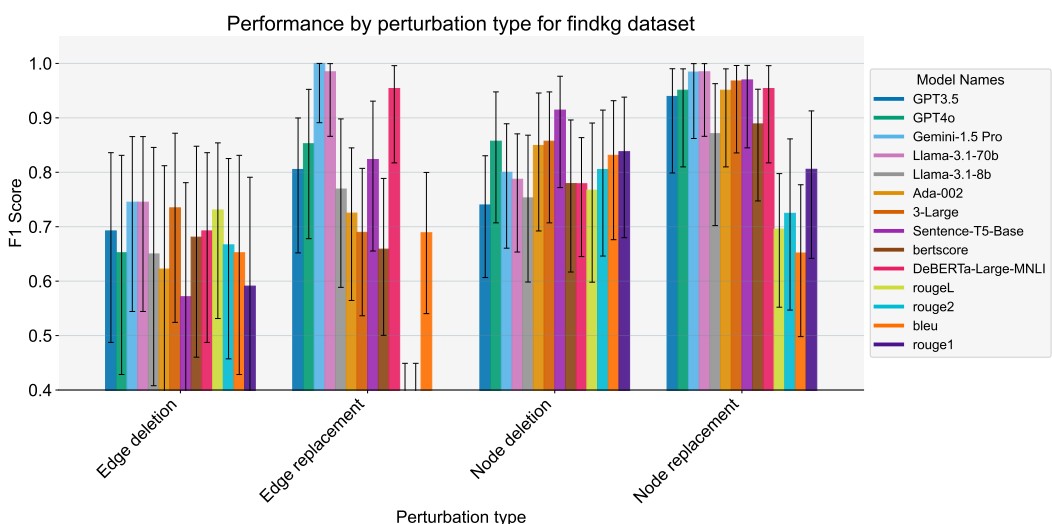

Figure 9: **Semantic Performance by Perturbation-Type for FinDKG Dataset.** Performance (F1 Score) of different semantic similarity models on the FinDKG dataset stratified by perturbation-type.

Table 11: Performance of different models on FINDKG dataset across perturbation types. Values shown as F1 score $\pm$ standard error.

|                      | Edge deletion      | Edge replacement   | Node deletion      | Node replacement   |
| -------------------- | ------------------ | ------------------ | ------------------ | ------------------ |
| 3-Large              | $0.735 \pm 0.089$  | $0.690 \pm 0.069$  | $0.857 \pm 0.061$  | $0.968 \pm 0.041$  |
| Ada-002              | $0.622 \pm 0.106$  | $0.725 \pm 0.072$  | $0.849 \pm 0.065$  | $0.951 \pm 0.046$  |
| DeBERTa-Large-MNLI   | $0.692 \pm 0.089$  | $0.954 \pm 0.046$  | $0.779 \pm 0.056$  | $0.954 \pm 0.046$  |
| GPT3.5               | $0.692 \pm 0.089$  | $0.805 \pm 0.063$  | $0.740 \pm 0.057$  | $0.939 \pm 0.049$  |
| GPT4o                | $0.652 \pm 0.103$  | $0.852 \pm 0.070$  | $0.857 \pm 0.061$  | $0.951 \pm 0.046$  |
| Gemini-1.5 Pro       | $0.745 \pm 0.082$  | $1.000 \pm 0.028$  | $0.800 \pm 0.058$  | $0.984 \pm 0.035$  |
| Llama-3.1-70b        | $0.745 \pm 0.082$  | $0.985 \pm 0.034$  | $0.787 \pm 0.055$  | $0.985 \pm 0.034$  |
| Llama-3.1-8b         | $0.650 \pm 0.112$  | $0.769 \pm 0.079$  | $0.753 \pm 0.069$  | $0.871 \pm 0.067$  |
| Sentence-T5-Base     | $0.571 \pm 0.112$  | $0.824 \pm 0.070$  | $0.914 \pm 0.052$  | $0.970 \pm 0.039$  |
| bertscore            | $0.681 \pm 0.099$  | $0.659 \pm 0.074$  | $0.779 \pm 0.071$  | $0.889 \pm 0.052$  |
| bleu                 | $0.652 \pm 0.103$  | $0.689 \pm 0.066$  | $0.831 \pm 0.065$  | $0.652 \pm 0.071$  |
| rouge1               | $0.591 \pm 0.109$  | $0.061 \pm 0.071$  | $0.838 \pm 0.066$  | $0.806 \pm 0.069$  |
| rouge2               | $0.667 \pm 0.094$  | $0.216 \pm 0.099$  | $0.805 \pm 0.068$  | $0.725 \pm 0.080$  |
| rougeL               | $0.731 \pm 0.082$  | $0.216 \pm 0.099$  | $0.767 \pm 0.075$  | $0.696 \pm 0.063$  |

## F.3   Globi Experimental Results

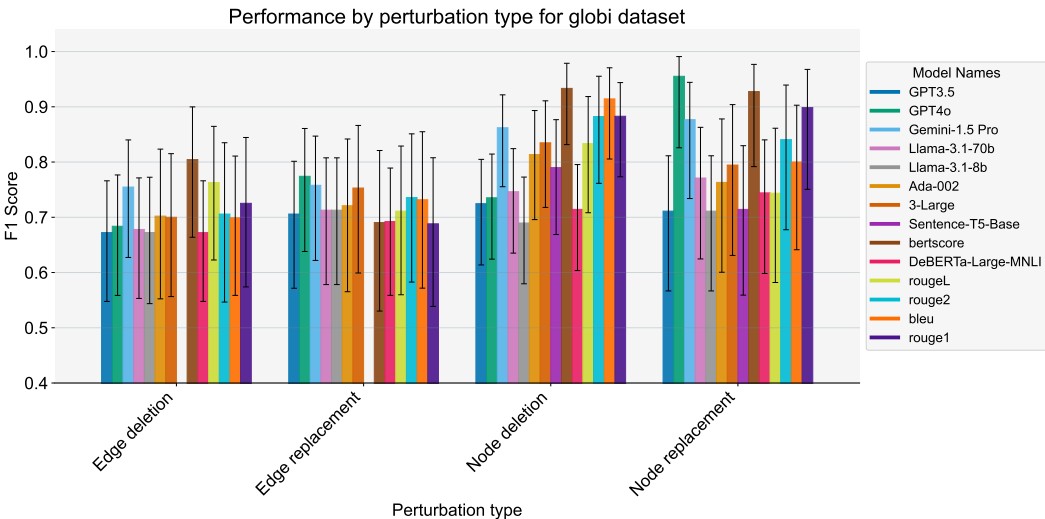

Figure 10: **Semantic Performance by Perturbation-Type for Globi Dataset.** Performance (F1 Score) of different semantic similarity models on the Globi dataset stratified by perturbation-type.

Table 12: Performance of different models on GLOBI dataset across perturbation types. Values shown as F1 score $\pm$ standard error.

|  | Edge deletion | Edge replacement | Node deletion | Node replacement |
|---|---|---|---|---|
| 3-Large | $0.700 \pm 0.066$ | $0.753 \pm 0.068$ | $0.835 \pm 0.049$ | $0.795 \pm 0.070$ |
| Ada-002 | $0.702 \pm 0.069$ | $0.721 \pm 0.071$ | $0.814 \pm 0.050$ | $0.763 \pm 0.071$ |
| DeBERTa-Large-MNLI | $0.672 \pm 0.056$ | $0.692 \pm 0.059$ | $0.714 \pm 0.049$ | $0.744 \pm 0.062$ |
| GPT3.5 | $0.672 \pm 0.056$ | $0.706 \pm 0.059$ | $0.725 \pm 0.049$ | $0.711 \pm 0.062$ |
| GPT4o | $0.684 \pm 0.056$ | $0.774 \pm 0.057$ | $0.735 \pm 0.049$ | $0.955 \pm 0.042$ |
| Gemini-1.5 Pro | $0.755 \pm 0.054$ | $0.758 \pm 0.057$ | $0.862 \pm 0.042$ | $0.877 \pm 0.054$ |
| Llama-3.1-70b | $0.678 \pm 0.056$ | $0.713 \pm 0.059$ | $0.746 \pm 0.048$ | $0.771 \pm 0.061$ |
| Llama-3.1-8b | $0.672 \pm 0.058$ | $0.713 \pm 0.059$ | $0.690 \pm 0.049$ | $0.711 \pm 0.062$ |
| Sentence-T5-Base | $0.095 \pm 0.071$ | $0.105 \pm 0.077$ | $0.790 \pm 0.053$ | $0.714 \pm 0.069$ |
| bertscore | $0.804 \pm 0.060$ | $0.690 \pm 0.074$ | $0.933 \pm 0.038$ | $0.928 \pm 0.047$ |
| bleu | $0.699 \pm 0.064$ | $0.732 \pm 0.072$ | $0.914 \pm 0.042$ | $0.800 \pm 0.067$ |
| rouge1 | $0.725 \pm 0.069$ | $0.688 \pm 0.069$ | $0.883 \pm 0.043$ | $0.899 \pm 0.055$ |
| rouge2 | $0.706 \pm 0.074$ | $0.736 \pm 0.068$ | $0.882 \pm 0.049$ | $0.841 \pm 0.067$ |
| rougeL | $0.763 \pm 0.062$ | $0.711 \pm 0.069$ | $0.833 \pm 0.054$ | $0.744 \pm 0.071$ |

## F.4 Oregano Experimental Results

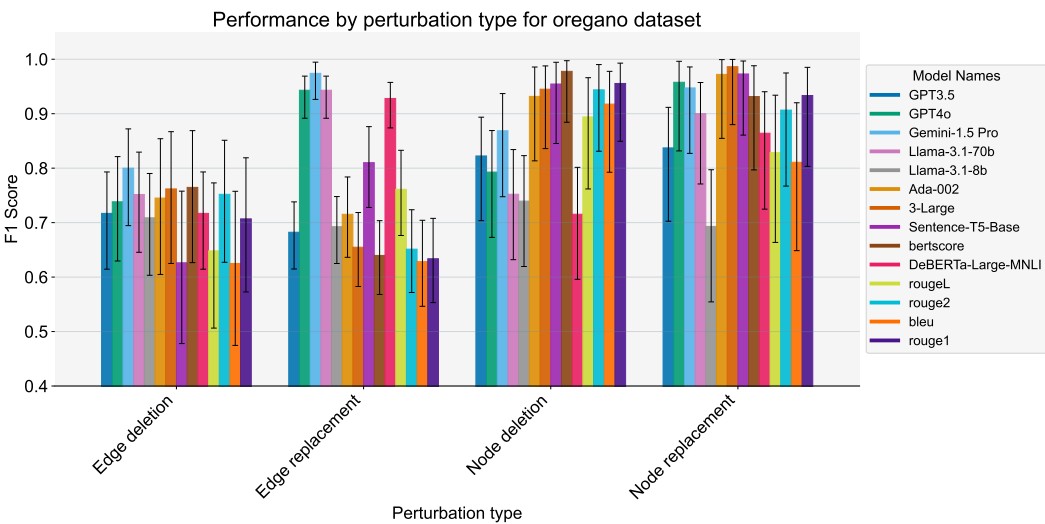

Figure 11: **Semantic Performance by Perturbation-Type for Oregano Dataset.** Performance (F1 Score) of different semantic similarity models on the Oregano dataset stratified by perturbation-type.

Table 13: Performance of different models on OREGANO dataset across perturbation types. Values shown as F1 score $\pm$ standard error.

|  | Edge deletion | Edge replacement | Node deletion | Node replacement |
|---|---|---|---|---|
| 3-Large | $0.762 \pm 0.062$ | $0.655 \pm 0.035$ | $0.945 \pm 0.039$ | $0.986 \pm 0.031$ |
| Ada-002 | $0.745 \pm 0.064$ | $0.715 \pm 0.038$ | $0.932 \pm 0.044$ | $0.972 \pm 0.037$ |
| DeBERTa-Large-MNLI | $0.717 \pm 0.046$ | $0.928 \pm 0.021$ | $0.715 \pm 0.052$ | $0.864 \pm 0.055$ |
| GPT3.5 | $0.717 \pm 0.046$ | $0.682 \pm 0.031$ | $0.822 \pm 0.048$ | $0.837 \pm 0.053$ |
| GPT4o | $0.738 \pm 0.049$ | $0.943 \pm 0.020$ | $0.793 \pm 0.050$ | $0.958 \pm 0.042$ |
| Gemini-1.5 Pro | $0.800 \pm 0.045$ | $0.974 \pm 0.017$ | $0.869 \pm 0.048$ | $0.947 \pm 0.040$ |
| Llama-3.1-70b | $0.752 \pm 0.047$ | $0.943 \pm 0.020$ | $0.752 \pm 0.052$ | $0.900 \pm 0.048$ |
| Llama-3.1-8b | $0.709 \pm 0.048$ | $0.693 \pm 0.031$ | $0.739 \pm 0.052$ | $0.693 \pm 0.062$ |
| Sentence-T5-Base | $0.626 \pm 0.071$ | $0.810 \pm 0.038$ | $0.955 \pm 0.038$ | $0.973 \pm 0.035$ |
| bertscore | $0.765 \pm 0.062$ | $0.640 \pm 0.035$ | $0.978 \pm 0.029$ | $0.932 \pm 0.049$ |
| bleu | $0.625 \pm 0.072$ | $0.628 \pm 0.040$ | $0.918 \pm 0.047$ | $0.811 \pm 0.069$ |
| rouge1 | $0.707 \pm 0.063$ | $0.634 \pm 0.039$ | $0.956 \pm 0.037$ | $0.933 \pm 0.046$ |
| rouge2 | $0.752 \pm 0.057$ | $0.651 \pm 0.039$ | $0.944 \pm 0.041$ | $0.907 \pm 0.053$ |
| rougeL | $0.648 \pm 0.068$ | $0.761 \pm 0.040$ | $0.894 \pm 0.052$ | $0.829 \pm 0.069$ |

## F.5 Statistical Analysis

We performed statistical analysis to assess the differences in model performance on our benchmark, accounting for the impact of perturbation-type and dataset. We analysed differences in F1 score using a linear mixed-effects model with dataset as a random-intercept and model and perturbation-type as fixed effects. Additionally, we modeled two-way interactions between model and dataset and model and perturbation-type. The full results are shown in table 14

Table 14: Statistical Analysis Results

| Model Summary | | | |
|---|---|---|---|
| No. Observations | 224 | No. Groups | 4 |
| Log-Likelihood | 73.032 | Scale (Residual Variance) | 0.0073 |

| Predictor | $\beta$ | p-value | Predictor | $\beta$ | p-value |
|---|---|---|---|---|---|
| Intercept | 0.848 | 0.000*** | Llama-3.1-8b × oregano | -0.061 | 0.653 |
| | | | | | |
| *Main Effect: Method Name* | | | Sentence-T5-Base × oregano | 0.002 | 0.989 |
| Ada-002 | -0.058 | 0.467 | bertscore × oregano | 0.008 | 0.955 |
| DeBERTa-Large-MNLI | -0.165 | 0.039* | bleu × oregano | -0.068 | 0.617 |
| GPT-3.5 | -0.145 | 0.070 | rouge1 × oregano | -0.002 | 0.987 |
| GPT-4o | -0.244 | 0.002** | rouge2 × oregano | 0.008 | 0.951 |
| Gemini-1.5 Pro | -0.136 | 0.088 | rougeL × oregano | -0.007 | 0.958 |
| | | | | | |
| Llama-3.1-70b | -0.136 | 0.088 | *Interaction: Method Name × Perturbation* | | |
| Llama-3.1-8b | -0.136 | 0.089 | Ada-002 × edge_replacement | 0.051 | 0.547 |
| Sentence-T5-Base | -0.217 | 0.007** | DeBERTa-Large-MNLI × edge_repl. | 0.153 | 0.073 |
| bertscore | -0.080 | 0.317 | GPT-3.5 × edge_replacement | 0.065 | 0.445 |
| bleu | -0.104 | 0.194 | GPT-4o × edge_replacement | 0.184 | 0.031* |
| rouge1 | -0.096 | 0.228 | Gemini-1.5 Pro × edge_repl. | 0.155 | 0.069 |
| rouge2 | -0.081 | 0.309 | Llama-3.1-70b × edge_repl. | 0.157 | 0.066 |
| rougeL | -0.059 | 0.461 | Llama-3.1-8b × edge_repl. | 0.071 | 0.407 |
| | | | | | |
| *Main Effect: Perturbation Type* | | | Sentence-T5-Base × edge_repl. | 0.102 | 0.232 |
| edge_replacement | -0.024 | 0.693 | bertscore × edge_replacement | -0.051 | 0.550 |
| node_removal | 0.124 | 0.039* | bleu × edge_replacement | 0.016 | 0.856 |
| node_replacement | 0.155 | 0.010** | rouge1 × edge_replacement | -0.142 | 0.096 |
| | | | | | |
| *Interaction: Method Name × Dataset* | | | rouge2 × edge_replacement | -0.113 | 0.185 |
| 3-Large × findkg | -0.100 | 0.461 | rougeL × edge_replacement | -0.114 | 0.182 |
| Ada-002 × findkg | -0.096 | 0.479 | Ada-002 × node_removal | 0.032 | 0.706 |
| DeBERTa-Large-MNLI × findkg | 0.067 | 0.620 | DeBERTa-Large-MNLI × node_remov. | -0.078 | 0.363 |
| GPT-3.5 × findkg | 0.016 | 0.907 | GPT-3.5 × node_removal | -0.044 | 0.610 |
| GPT-4o × findkg | 0.084 | 0.532 | GPT-4o × node_removal | -0.010 | 0.903 |
| Gemini-1.5 Pro × findkg | 0.064 | 0.638 | Gemini-1.5 Pro × node_remov. | -0.026 | 0.759 |
| Llama-3.1-70b × findkg | 0.066 | 0.624 | Llama-3.1-70b × node_remov. | -0.059 | 0.490 |
| Llama-3.1-8b × findkg | -0.008 | 0.951 | Llama-3.1-8b × node_remov. | -0.066 | 0.441 |
| Sentence-T5-Base × findkg | -0.019 | 0.885 | Sentence-T5-Base × node_remov. | 0.260 | 0.002** |
| bertscore × findkg | -0.069 | 0.610 | bertscore × node_removal | 0.015 | 0.864 |
| bleu × findkg | -0.107 | 0.428 | bleu × node_removal | 0.086 | 0.315 |
| rouge1 × findkg | -0.236 | 0.081 | rouge1 × node_removal | 0.089 | 0.297 |
| rouge2 × findkg | -0.202 | 0.135 | rouge2 × node_removal | 0.061 | 0.477 |
| rougeL × findkg | -0.188 | 0.165 | rougeL × node_removal | -0.006 | 0.944 |
| 3-Large × globi | -0.141 | 0.295 | Ada-002 × node_replacement | 0.031 | 0.721 |
| Ada-002 × globi | -0.132 | 0.327 | DeBERTa-Large-MNLI × node_repl. | 0.047 | 0.584 |
| DeBERTa-Large-MNLI × globi | -0.072 | 0.594 | GPT-3.5 × node_replacement | 0.024 | 0.777 |
| GPT-3.5 × globi | -0.075 | 0.579 | GPT-4o × node_replacement | 0.129 | 0.130 |
| GPT-4o × globi | 0.043 | 0.749 | Gemini-1.5 Pro × node_repl. | 0.043 | 0.612 |
| Gemini-1.5 Pro × globi | -0.006 | 0.965 | Llama-3.1-70b × node_repl. | 0.036 | 0.670 |
| Llama-3.1-70b × globi | -0.082 | 0.543 | Llama-3.1-8b × node_repl. | -0.032 | 0.706 |
| Llama-3.1-8b × globi | -0.073 | 0.591 | Sentence-T5-Base × node_repl. | 0.216 | 0.011* |
| Sentence-T5-Base × globi | -0.413 | 0.002** | bertscore × node_replacement | -0.008 | 0.924 |
| bertscore × globi | 0.018 | 0.893 | bleu × node_replacement | -0.082 | 0.337 |
| bleu × globi | -0.027 | 0.843 | rouge1 × node_replacement | 0.030 | 0.728 |
| rouge1 × globi | -0.011 | 0.936 | rouge2 × node_replacement | -0.049 | 0.565 |
| rouge2 × globi | -0.014 | 0.917 | rougeL × node_replacement | -0.132 | 0.122 |
| rougeL × globi | -0.027 | 0.839 | | | |
| 3-Large × oregano | -0.075 | 0.579 | | | |
| Ada-002 × oregano | -0.041 | 0.760 | | | |
| DeBERTa-Large-MNLI × oregano | 0.028 | 0.833 | | | |
| GPT-3.5 × oregano | -0.014 | 0.919 | | | |
| GPT-4o × oregano | 0.114 | 0.398 | | | |
| Gemini-1.5 Pro × oregano | 0.079 | 0.560 | | | |
| Llama-3.1-70b × oregano | 0.028 | 0.838 | | | |

*Note*: $\beta$ is the coefficient. *p <0.05, **p <0.01, ***p <0.001.

# G    Dataset Licenses

All datasets used in this manuscript are free and open-source. Below we describe the licenses associated with each.

**Codex** is licensed under an MIT license. WikiData, which Codex is based upon, is licensed under a CC0 License.

**FinDKG** is made freely available for research purposes (non-commercial purposes only).

**Globi** is licensed under a CC by 4.0 license.

**Oregano** is licensed under a CC by 4.0 license.

# H    Edge-Replacement Mappings

For all datasets a custom-mapping was defined indicating what value an edge could be replaced with for an edge-replacement perturbation 2.2. Below we describe the mapping used for each dataset, where the list of values for a given key indicates allowed replacement values for that edge.

**Codex**

```
{
    "cast member": [
        "creator",
        "director"
    ],
    "director": [
        "cast member",
        "creator"
    ],
    "creator": [
        "cast member",
        "director"
    ],
    "author": [
        "director",
        "cast member"
    ],
    "headquarters location": [
        "capital"
    ],
    "located in the administrative terroritorial entity": [
        "shares border with",
        "diplomatic relation"
    ],
    "country": [
        "shares border with",
        "diplomatic relation"
    ],
    "shares border with": [
        "located in the administrative terroritorial entity",
        "named after",
        "country",
        "diplomatic relation"
    ],
    "diplomatic relation": [
        "located in the administrative terroritorial entity",
        "shares border with",
        "country",
        "part of"
```

```
    ],
    "location of formation": [
        "headquarters location"
    ],
    "country of origin": [
        "narrative location"
    ],
    "chairperson": [
        "founded by",
        "chief executive officer",
        "director"
    ],
    "parent organization": [
        "founded by",
        "part of"
    ],
    "airline hub": [
        "headquarters location"
    ],
    "chief executive officer": [
        "founded by",
        "chairperson",
        "director"
    ],
    "founded by": [
        "chairperson",
        "director",
        "chief executive officer"
    ],
    "airline alliance": [
        "member of"
    ],
    "narrative location": [
        "country of origin"
    ],
    "architect": [
        "named after"
    ],
    "capital": [
        "headquarters location"
    ],
    "country of citizenship": [
        "place of burial",
        "place of birth",
        "place of death"
    ],
    "residence": [
        "place of death",
        "place of birth",
        "place of burial"
    ],
    "place of birth": [
        "place of death",
        "place of burial",
        "residence"
    ],
    "place of death": [
        "place of birth",
        "place of burial",
```

```
            "residence"
        ],
        "place of burial": [
            "place of birth",
            "place of death",
            "residence"
        ],
        "child": [
            "sibling",
            "spouse",
            "unmarried partner",
            "influenced by"
        ],
        "spouse": [
            "sibling",
            "influenced by",
            "child",
            "unmarried partner"
        ],
        "sibling": [
            "child",
            "spouse",
            "unmarried partner",
            "influenced by"
        ],
        "unmarried partner": [
            "sibling",
            "spouse",
            "influenced by",
            "child"
        ],
        "educated at": [
            "employer"
        ],
        "cause of death": [
            "medical condition",
            "health specialty"
        ],
        "medical condition": [
            "cause of death",
            "health specialty"
        ],
        "heath specialty": [
            "medical condition",
            "cause of death"
        ],
        "member of political party": [
            "employer"
        ],
        "publisher": [
            "record label",
            "employer"
        ],
        "record label": [
            "publisher",
            "employer"
        ]
    }
```

**FinDKG**

```
{
    "Positive_Impact_On": [
        "Negative_Impact_On"
    ],
    "Negative_Impact_On": [
        "Positive_Impact_On"
    ],
    "Raise": [
        "Decrease"
    ],
    "Decrease": [
        "Raise"
    ]
}
```

**Globi**

```
{
    "parasiteOf": [
        "eats",
        "commensalistOf",
        "mutualistOf"
    ],
    "hasHost": [
        "coOccursWith"
    ],
    "adjacentTo": [
        "hasHost",
        "preysOn"
    ],
    "coOccursWith": [
        "hasHost"
    ],
    "visitsFlowersOf": [
        "pathogenOf"
    ],
    "preysOn": [
        "livesNear"
    ],
    "pollinates": [
        "eats",
        "visits"
    ],
    "kills": [
        "coOccursWith",
        "pathogenOf"
    ],
    "rootparasiteOf": [
        "adjacentTo",
        "livesNear",
        "eats",
        "commensalistOf",
        "mutualistOf"
    ],
    "hasVector": [
        "pathogenOf"
    ],
    "pathogenOf": [
```

```
            "visitsFlowersOf"
        ],
        "mutualistOf": [
            "visits",
            "eats",
            "hasHost",
            "parasiteOf"
        ],
        "livesInsideOf": [
            "eats",
            "adjacentTo"
        ],
        "livesUnder": [
            "eats",
            "visitsFlowersOf",
            "visits",
            "pollinates"
        ],
        "epiphyteOf": [
            "hasHabitat",
            "parasiteOf",
            "adjacentTo",
            "livesNear"
        ],
        "inhabits": [
            "pathogenOf"
        ],
        "ectoparasiteOf": [
            "eats",
            "commensalistOf",
            "mutualistOf"
        ],
        "endoparasiteOf": [
            "commensalistOf",
            "mutualistOf"
        ],
        "kleptoparasiteOf": [
            "eats",
            "coOccursWith",
            "commensalistOf",
            "mutualistOf"
        ],
        "providesNutrientsFor": [
            "eats"
        ],
        "hasDispersalVector": [
            "eats"
        ],
        "endoparasitoidOf": [
            "eats"
        ],
        "guestOf": [
            "eats",
            "coOccursWith",
            "preysOn",
            "visitsFlowersOf"
        ],
        "livesNear": [
            "parasiteOf"
```

```
        ] ,
        "hasRoost ": [
            "pathogenOf",
            "preysOn",
            "hasHost",
            "eats"
        ] ,
        "coRoostsWith ": [
            "pathogenOf",
            "preysOn",
            "hasHost",
            "eats"
        ] ,
        "ectoParasitoid ": [
            "eats"
        ] ,
        "allelopathOf ": [
            "pathogenOf",
            "hasHost",
            "adjacentTo"
        ]
}
```

**Oregano**

```
{
    "increase_activity ": [
        "decrease_activity"
    ] ,
    "decrease_activity ": [
        "increase_activity"
    ] ,
    "increase_effect ": [
        "decrease_effect"
    ] ,
    "decrease_effect ": [
        "increase_effect"
    ]
}
```

# I  Prompt Templates

## I.1  LLM-as-a-judge Evaluation Prompt Template

For the experimental results reported in 6.2 we used the following prompt template for all LLM-as-a-judge models:

```
You will be provided with two biomedical statements.

Your goal is to determine whether these statements are
    semantically similar or not.

If the statements describe different concepts or different
    relationships between these concepts they are not semantically
     similar, even if there is overlap in the words used to
    describe them.

Please reply with only a \"yes\" if the pieces of text are
    semantically similar, or \"no\" if they are not.
```

```
Statement 1: {statement_1}
Statement 2: {statement_2}
```

Your answer:

## I.2 Response Generation Prompt Templates

For response generation we used a similar template for all datasets, adapting the few-shot examples on a per-dataset basis. Below we report the generation templates used for every dataset:

**Codex**

```
You are going to be given a list of triples from a directed
    knowledge graph. Each triple consists of a subject, a relation
    , and an object.

The triples are defined in directed order, where the relationship
    direction is from "source_node" to "target_node"

Your goal is to express this triple in a continuous natural
    language statement suitable for a general or a scientific
    audience.

For example, given the triple:

triple = [
    {{'source_node': {{'name': 'Norway'}}, 'relation': {{'name': '
        diplomatic relation'}}, 'target_node': {{'name': 'Colombia
        '}}}},
    {{'source_node': {{'name': 'Norway'}}, 'relation': {{'name': '
        shares border with'}}, 'target_node': {{'name': 'Russia
        '}}}},
    {{'source_node': {{'name': 'Norway'}}, 'relation': {{'name': '
        member of'}}, 'target_node': {{'name': 'Organisation for
        the Prohibition of Chemical Weapons'}}}},
    {{'source_node': {{'name': 'Russia'}}, 'relation': {{'name': '
        diplomatic relation'}}, 'target_node': {{'name': 'Ethiopia
        '}}}},
    {{'source_node': {{'name': 'Colombia'}}, 'relation': {{'name':
        'member of'}}, 'target_node': {{'name': 'Universal Postal
        Union'}}}},
    {{'source_node': {{'name': 'Colombia'}}, 'relation': {{'name':
        'member of'}}, 'target_node': {{'name': 'Andean Community
        '}}}},
    {{'source_node': {{'name': 'Colombia'}}, 'relation': {{'name':
        'member of'}}, 'target_node': {{'name': 'Organisation for
        the Prohibition of Chemical Weapons'}}}},
]

You could say:
"Norway has diplomatic relations with Colombia and Russia. Russia
    is also known to have diplomatic relations with Ethiopia. Both
     Norway and Colombia are members of the Organisation for the
    Prohibition of Chemical weapons, while Colombia is also a
    member of the Universal Postal Union and the Andean Community
    ."

or given the triple:
```

```
triple = [
    {{'source_node': {{'name': 'Renate Axt'}}, 'relation': {{'name
        ': 'occupation'}}, 'target_node': {{'name': 'writer'}}}},
    {{'source_node': {{'name': 'Renate Axt'}}, 'relation': {{'name
        ': 'place of birth'}}, 'target_node': {{'name': 'Darmstadt
        '}}}},
    {{'source_node': {{'name': 'Darmstadt'}}, 'relation': {{'name
        ': 'country'}}, 'target_node': {{'name': 'Germany'}}}}
    {{'source_node': {{'name': 'Germany'}}, 'relation': {{'name':
        'shares border with'}}, 'target_node': {{'name': 'Austria
        '}}}},
    {{'source_node': {{'name': 'Germany'}}, 'relation': {{'name':
        'shares border with'}}, 'target_node': {{'name': '
        Czechoslovakia'}}}},
]
```

You could say:
"Renate Axt was a writer, born in Darmstadt in Germany. Germany
    shares a border with Austria and Czechoslovakia."

or given the triple:

```
triple = [
    {{'source_node': {{'name': 'Frederick William II of Prussia
        '}}, 'relation': {{'name': 'child'}}, 'target_node': {{'
        name': 'Friedrich Wilhelm, Count Brandenburg'}}}},
    {{'source_node': {{'name': 'Frederick William II of Prussia
        '}}, 'relation': {{'name': 'member of'}}, 'target_node':
        {{'name': 'Saint Petersburg Academy of Sciences'}}}},
    {{'source_node': {{'name': 'Friedrich Wilhelm, Count
        Brandenburg'}}, 'relation': {{'name': 'place of birth'}},
        'target_node': {{'name': 'Berlin'}}}},
    {{'source_node': {{'name': 'Friedrich Wilhelm, Count
        Brandenburg'}}, 'relation': {{'name': 'occupation'}}, '
        target_node': {{'name': 'politician'}}}},
    {{'source_node': {{'name': 'Saint Petersburg Academy of
        Sciences'}}, 'relation': {{'name': 'country'}}, '
        target_node': {{'name': 'Russia'}}}},
    {{'source_node': {{'name': 'Saint Petersburg Academy of
        Sciences'}}, 'relation': {{'name': 'located in the
        administrative terroritorial entity'}}, 'target_node': {{'
        name': 'Saint Petersburg'}}}},
]
```

You could say:
"Frederick William II of Prussia was the child of Friedrich
    Wilhelm, Count Brandenburg, a politician who was born in
    Berlin. Frederick William II was a member of the Saint
    Petersburg Academy of Sciences, a Russian institute, located
    in Saint Petersburg"

Rules:

You must use all of the entities provided in the triple and please
    include each node verbatim but do not use quotes.
```

Your statement must preserve the directions of the knowledge−graph
    .

Do NOT list the items in the triple as a list. Instead, write a
    sentence or paragraph that describes the relationship between
    every item in the triple.

You can also add additional information to the triple to make the
    relationship more clear, however you must include all the
    triples in your response.

In your final response, do NOT put name of any node or relation in
    quotes.
For example for the node '{{"name": "The Godfather"}}':
    'The movie "The Godfather"...' is **not** allowed
    '... starred in the movie "The Godfather"' is **not** allowed
    '... alongside Marlon Brando in 'The Godfather' is **not**
        allowed

You may slightly rephrase the names of edges to ensure they are
    grammatically correct and produce a fluent, coherent sounding
    statement

Triples: {triples}

**FinDKG**

For FinDKG we provided a definition for every edge value due to the ambiguity in the edge name
alone.

You are going to be given a list of triples from a directed
    knowledge graph. Each triple consists of a subject, a relation
    , and an object.

The triples are defined in directed order, where the relationship
    direction is from "source_node" to "target_node"

Your goal is to express this triple in a continuous natural
    language statement suitable for a general or a scientific
    audience.

For example, given the triple:

triple = [
    {{'source_node': {{'name': 'U.S. Air Force'}}, 'relation': {{'
        name': 'Control'}}, 'target_node': {{'name': 'Asia and
        Europe'}}}},
    {{'source_node': {{'name': 'U.S. Air Force'}}, 'relation': {{'
        name': 'Operate_In'}}, 'target_node': {{'name': '
        Afghanistan'}}}},
    {{'source_node': {{'name': 'Afghanistan'}}, 'relation': {{'
        name': 'Has'}}, 'target_node': {{'name': 'government'}}}},
]

You could say:
"The U.S. Air Force controls Asia and Europe, and operates in
    Afghanistan. Afghanistan has a government."

or given the triple:

```
triple = [
    {{'source_node': {{'name': 'Italian Debt'}}, 'relation': {{'
        name': 'Impact'}}, 'target_node': {{'name': 'Investors
        '}}}},
    {{'source_node': {{'name': 'Italian Debt'}}, 'relation': {{'
        name': 'Impact'}}, 'target_node': {{'name': 'Italian
        Government'}}}},
    {{'source_node': {{'name': 'Italian Debt'}}, 'relation': {{'
        name': 'Relate_To'}}, 'target_node': {{'name': 'European
        Central Bank'}}}},
    {{'source_node': {{'name': 'Investors'}}, 'relation': {{'name
        ': 'Impact'}}, 'target_node': {{'name': 'Yuan'}}}},
]
```

You could say:
"Italian debt is related to policies of the European Central Bank.
    This debt has an impact on the Italian Government in addition
    to investors. Investors may also subsequently impact the
    value of Yuan."

or given the triple:

```
triple = [
    {{'source_node': {{'name': 'Federal Reserve System'}}, '
        relation': {{'name': 'Impact'}}, 'target_node': {{'name':
        'Gold'}}}},
    {{'source_node': {{'name': 'Federal Reserve System'}}, '
        relation': {{'name': 'Control'}}, 'target_node': {{'name':
        'Expenses'}}}},
    {{'source_node': {{'name': 'Federal Reserve System'}}, '
        relation': {{'name': 'Control'}}, 'target_node': {{'name':
        'The U.S. Economy'}}}},
    {{'source_node': {{'name': 'The U.S. Economy'}}, 'relation':
        {{'name': 'Relate_To'}}, 'target_node': {{'name': 'Gross
        Domestic Product'}}}},
    {{'source_node': {{'name': 'The U.S. Economy'}}, 'relation':
        {{'name': 'Relate_To'}}, 'target_node': {{'name': 'U.S.
        stocks'}}}},
    {{'source_node': {{'name': 'Expenses'}}, 'relation': {{'name':
        'Positive_Impact_On'}}, 'target_node': {{'name': 'Gold
        '}}}},
]
```

You could say:
"The Federal Reserve System controls expenses, which can have a
    positive impact on Gold, an asset also impacted by the Federal
    Reserve System. Additionally this system controls the U.S.
    Economy which has a relationship with Gross Domestic Product
    and U.S. stocks."

Rules:

You must use all of the entities provided in the triple and please
    include each node verbatim but do not use quotes.

Your statement must preserve the directions of the knowledge−graph
.

Do NOT list the items in the triple as a list. Instead, write a
sentence or paragraph that describes the relationship between
every item in the triple.

You can also add additional information to the triple to make the
relationship more clear, however you must include all the
triples in your response.

In your final response, do NOT put name of any node or relation in
quotes.
For example for the node '{{"name": "Federal Reserve System"}}':
'The "Federal Reserve System"...' is **not** allowed
'... is controlled by the "Federal Reserve System"' is **not**
allowed
'... is controlled by the 'Federal Reserve System'' is **not**
allowed

Relationship Definitions and Examples:
− Has: Indicates ownership or possession, often of assets or
subsidiaries in a financial context. Example: Google Has
Android.
− Announce: Refers to the formal public declaration of a financial
event, product launch, or strategic move. Example: Apple
Announces iPhone 13.
− Operate_In: Describes the geographical market in which a
business entity conducts its operations. Example: Tesla
Operates In China.
− Introduce: Denotes the first−time introduction of a financial
instrument, product, or policy to the market. Example: Samsung
Introduces Foldable Screen.
− Produce: Specifies the entity responsible for creating a
particular product, often in a manufacturing or financial
product context. Example: Pfizer Produces Covid−19 Vaccine.
− Control: Implies authority or regulatory power over monetary
policy, financial instruments, or market conditions. Example:
Federal Reserve Controls Interest Rates.
− Participates_In: Indicates active involvement in an event that
has financial or economic implications. Example: United States
Participates In G20 Summit.
− Impact: Signifies a notable effect, either positive or negative,
on market trends, financial conditions, or economic
indicators. Example: Brexit Impacts European Union.
− Positive_Impact_On: Highlights a beneficial effect on financial
markets, economic indicators, or business performance. Example
: Solar Energy Positive Impact On ESG Ratings.
− Negative_Impact_On: Underlines a detrimental effect on financial
markets, economic indicators, or business performance.
Example: Covid−19 Negative Impact On Tourism Sector.
− Relate_To: Points out a connection or correlation with a
financial concept, sector, or market trend. Example: AI
Relates To FinTech Sector.
− Is_Member_Of: Denotes membership in a trade group, economic
union, or financial consortium. Example: Germany Is Member Of
EU.

- Invests_In: Specifies an allocation of capital into a financial
  instrument, sector, or business entity. Example: Warren
  Buffett Invests In Apple.
- Raise: Indicates an increase, often referring to capital,
  interest rates, or production levels in a financial context.
  Example: OPEC Raises Oil Production.
- Decrease: Indicates a reduction, often referring to capital,
  interest rates, or production levels in a financial context.
  Example: Federal Reserve Decreases Interest Rates.

Triples: {triples}

**Globi**

You are going to be given a list of triples from a directed
knowledge graph. Each triple consists of a subject, a relation
, and an object.

The triples are defined in directed order, where the relationship
direction is from "source_node" to "target_node"

Your goal is to express this triple in a continuous natural
language statement suitable for a general or a scientific
audience.

For example, given the triple:

```
triple = [
    {{'source_node': {{'name': 'Pinus jeffreyi'}}, 'relation': {{'
        name': 'interactsWith'}}, 'target_node': {{'name': 'Betula
        occidentalis'}}}},
    {{'source_node': {{'name': 'Pinus jeffreyi'}}, 'relation': {{'
        name': 'interactsWith'}}, 'target_node': {{'name': '
        Wyethia mollis'}}}},
    {{'source_node': {{'name': 'Wyethia mollis'}}, 'relation': {{'
        name': 'interactsWith'}}, 'target_node': {{'name': '
        Collomia heterophylla'}}}}
]
```

You could say:
"Pinus jeffreyi interacts with several species including Betula
occidentalis and Wyethia mollis. Wyethia mollis in turn
interacts with Collomia heterophylla"

```
triple = [
    {{'source_node': {{'name': 'Neotoma mexicana'}}, 'relation':
        {{'name': 'interactsWith'}}, 'target_node': {{'name': '
        Lynx rufus'}}}},
    {{'source_node': {{'name': 'Neotoma mexicana'}}, 'relation':
        {{'name': 'interactsWith'}}, 'target_node': {{'name': '
        Canis latrans'}}}}
    {{'source_node': {{'name': 'Canis latrans'}}, 'relation': {{'
        name': 'eats'}}, 'target_node': {{'name': 'Prunus serotina
        '}}}},
    {{'source_node': {{'name': 'Canis latrans'}}, 'relation': {{'
        name': 'eats'}}, 'target_node': {{'name': 'Sylvilagus
        cunicularius'}}}},
```

```
        {{'source_node': {{'name': 'Canis latrans'}}, 'relation': {{'
            name': 'coOccursWith'}}, 'target_node': {{'name': '
            Panthera leo'}}}},
        {{'source_node': {{'name': 'Crotalus pricei'}}, 'relation':
            {{'name': 'preysOn'}}, 'target_node': {{'name': '
            Sceloporus jarrovii'}}}},
        {{'source_node': {{'name': 'Crotalus pricei'}}, 'relation':
            {{'name': 'preysOn'}}, 'target_node': {{'name': 'Neotoma
            mexicana'}}}},
        {{'source_node': {{'name': 'Crotalus pricei'}}, 'relation':
            {{'name': 'preysOn'}}, 'target_node': {{'name': 'Junco
            phaeonotus'}}}},
]
```

You could say:
"Crotalus pricei is a species known to prey on several species
    including, Sceloporus jarrovii, Junco phaeonotus, and Neotoma
    mexicana. Neotoma mexicana have interactions with several
    species including Lynx rufus and Canis latrans. Canis latrans
    are known to co−occur with Panthera leo and eat Prunus
    serotina and Sylvilagus cuncicularius."

```
triple = [
    {{'source_node': {{'name': 'Chromatomyia erigerontophaga'}}, '
        relation': {{'name': 'visitsFlowersOf'}}, 'target_node':
        {{'name': 'Potentilla nivea'}}}},
    {{'source_node': {{'name': 'Chromatomyia erigerontophaga'}}, '
        relation': {{'name': 'pollinates'}}, 'target_node': {{'
        name': 'Erigeron compositus'}}}},
    {{'source_node': {{'name': 'Potentilla nivea'}}, 'relation':
        {{'name': 'interactsWith'}}, 'target_node': {{'name': '
        Salix arctica'}}}},
    {{'source_node': {{'name': 'Potentilla nivea'}}, 'relation':
        {{'name': 'interactsWith'}}, 'target_node': {{'name': '
        Draba nivalis'}}}},
    {{'source_node': {{'name': 'Erigeron compositus'}}, 'relation
        ': {{'name': 'interactsWith'}}, 'target_node': {{'name': '
        Populus tremuloides'}}}},
    {{'source_node': {{'name': 'Erigeron compositus'}}, 'relation
        ': {{'name': 'interactsWith'}}, 'target_node': {{'name': '
        Luetkea pectinata'}}}}
]
```

You could say:
"Chromatomyia erigerontophaga is a species known to visit the
    flowers of Potentilla nivea which in turn have interactions
    with Salix arctica and Draba nivalis. C. erigerontophaga also
    pollinate Erigeron compositus. This flower is known to
    interact with Populus tremuloides and Luetkea pectinata."

Rules:

You must use all of the entities provided in the triple and please
    include each node verbatim but do not use quotes.

Your statement must preserve the directions of the knowledge−graph
    .
```

Do NOT list the items in the triple as a list. Instead, write a sentence or paragraph that describes the relationship between every item in the triple.

You can also add additional information to the triple to make the relationship more clear, however you must include all the triples in your response.

In your final response, do NOT put name of any node or relation in quotes.

For example for the node '{{"name": "Baccharis sarothroides"}}':
    'The "Baccharis sarothroides"...' is **not** allowed
    '... interacts with the plant "Baccharis sarothroides"' is **not** allowed
    "The plant 'Baccharis sarothroides'..." is **not** allowed

Triples: {triples}

**Oregano**

You are going to be given a list of triples from a directed knowledge graph. Each triple consists of a subject, a relation, and an object.

The triples are defined in directed order, where the relationship direction is from "source_node" to "target_node"

Your goal is to express this triple in a continuous natural language statement suitable for a general or a scientific audience.

For example, given the triple:

```
triple = [
    {{'source_node': {{'name': 'GFRA2'}}, 'relation': {{'name': '
        acts_within'}}, 'target_node': {{'name': 'NCAM1
        interactions'}}}},
    {{'source_node': {{'name': 'GFRA2'}}, 'relation': {{'name': '
        acts_within'}}, 'target_node': {{'name': 'RAF/MAP kinase
        cascade'}}}},
    {{'source_node': {{'name': 'GFRA2'}}, 'relation': {{'name': '
        acts_within'}}, 'target_node': {{'name': 'RET signaling
        '}}}}
]
```

You could say:
"The gene GRFA2 acts within several pathways including NCAM1 interactions pathway, RAF/MAP kinase cascade and RET signaling.

or given the triple:

```
triple = [
    {{'source_node': {{'name': 'nephrotic syndrome'}}, 'relation':
        {{'name': 'has_phenotype'}}, 'target_node': {{'name': '
        microcephaly'}}}},
```

```
        {{'source_node': {{'name': 'SGPL1'}}, 'relation': {{'name': '
            causes_condition'}}, 'target_node': {{'name': 'nephrotic
            syndrome'}}}},
        {{'source_node': {{'name': 'SGPL1'}}, 'relation': {{'name': '
            acts_within'}}, 'target_node': {{'name': 'Sphingolipid de
            novo biosynthesis'}}}}
]
```

You could say:
"The gene SGPL1 which acts within the sphingolipid de novo
    biosynthesis pathway and causes nephrotic syndrome, a disease
    characterised by symptoms such as microcephaly"

or given the triple:

```
triple = [
    {{'source_node': {{'name': 'methyl 1-phenylalaninate'}}, '
        relation': {{'name': 'has_target'}}, 'target_node': {{'
        name': 'fimbrial protein'}}}},
    {{'source_node': {{'name': 'methyl 1-phenylalaninate'}}, '
        relation': {{'name': 'has_target'}}, 'target_node': {{'
        name': 'prothrombin'}}}},
    {{'source_node': {{'name': 'prothrombin'}}, 'relation': {{'
        name': 'gene_product_of'}}, 'target_node': {{'name': 'F2
        '}}}},
    {{'source_node': {{'name': 'F2'}}, 'relation': {{'name': '
        causes_condition'}}, 'target_node': {{'name': 'pregnancy
        loss'}}}}
]
```

You could say:
"The drug methyl 1-phenylalaninate targets both the fimbrial
    protein and prothrombin. Prothrombin is a gene product of F2
    which is known to cause pregnancy loss.

Rules:

You must use all of the entities provided in the triple and please
    include each node verbatim but do not use quotes.

Your statement must preserve the directions of the knowledge-graph
    .

Do NOT list the items in the triple as a list. Instead, write a
    sentence or paragraph that describes the relationship between
    every item in the triple.

You can also add additional information to the triple to make the
    relationship more clear, however you must include all the
    triples in your response.

In your final response, do NOT put name of any node or relation in
    quotes.
For example for the node '{{"name": "2-(6-methylpyridin-2-yl)-n-
    pyridin-4-ylquinazolin-4-amine"}}':
    'The drug "2-(6-methylpyridin-2-yl)-n-pyridin-4-ylquinazolin
        -4-amine"...' is **not** allowed

```
'... is activate by the drug "2−(6−methylpyridin −2−yl)−n−
    pyridin −4−ylquinazolin −4−amine"' is **not** allowed
'... is activate by the drug 2−(6−methylpyridin −2−yl)−n−
    pyridin −4−ylquinazolin −4−amine' is allowed
```

Ensure you correctly capitalize the entity in your statement. The
    capitalization does not need to match the provided entity.
For example for the node: '{{"name": "Common Pathway of Fibrin
    Clot Formation"}}':

```
'...acts within the Common Pathway of Fibrin Clot Formation
    ...' is **not** correct capitalization
'...acts within the common pathway of fibrin clot formation
    ...' is correct capitalization
```

You may slightly rephrase the names of pathways to ensure they are
    grammatically correct in your generated statement.

Triples: {triples}

### I.3   Response Validation Prompt Templates

As with response generation, we used a similar prompt for every dataset, adapting the fewshot
example accordingly. The prompts used for both entity-extraction and KG-extraction for each dataset
are shown below.

**Codex - Entity Extraction Prompt Template**

You will be provided with a statement describing the relationship
    between various entities.

The different entities described are of the following types: ['
    complication of pregnancy, childbirth and the puerperium', '
    occupation', 'manga', 'systemic disease', 'climbing', '
    government agency', 'paramilitary', 'type of legal entity in
    Germany', 'superior planet', 'county seat', 'legislative term
    ', 'clergy', 'region of France', 'movement in cinema', '
    political organisation', 'supervillain', 'tapestry', 'rock
    band', 'religious servant', 'constructed language', 'military
    alliance', 'poetry collection', 'botanical garden', 'former
    municipality of Norway', 'episcopal see', 'municipality of
    Germany', 'language family', 'film character', 'city of
    Indonesia', 'association under the French law of 1901', '
    county of Florida', 'township of New Jersey', 'regional
    organization', 'auto racing team', 'person', 'digital library
    ', 'tribute album', 'media franchise', 'film school', '
    Australian rules football league', 'lake', 'national academy',
    'imperial abbey', 'Wikimedia list of persons', 'locality', '
    kingdom', 'Comune sparso', 'Parisian cemetery', 'municipality
    of Tunisia', 'higher municipal association of Germany', '
    necked bowl lutes', 'autonomous soviet socialist republic of a
    union republic of the Soviet Union', 'realm', 'medical
    specialist', 'Christian Church', 'parliamentary assembly', '
    frazione', 'island nation', 'United States national laboratory
    ', 'UCI ProTeam', 'chemical substance', 'academic', '
    independent city', 'parliamentary term in the Kingdom of Great
    Britain', 'website', 'municipality of the Netherlands', '
    sultanate', 'club', 'university−preparatory school', 'United
    States federal agency', 'comics character', 'set of wall
    hangings', 'personal union', 'human biblical figure', 'Islam',

'airline alliance ', 'geopolitical community ', 'federal state ', 'Field Operating Agency ', 'television genre ', 'minor party ', 'esports league ', 'art movement ', 'branch of biology ', 'municipality of Brazil ', 'ethnic community ', 'definitely endangered language ', 'research object ', 'writer ', 'cleric ', 'necked box lutes played with a bow ', 'stadium ', 'state of Malaysia ', 'Parliamentary group ', 'cargo airline ', 'law ', 'Conservatory of regional relevance ', 'district of Canton Thurgau ', 'think tank ', 'district capital ', 'major basilica ', 'political party in the Russian Empire ', 'city of rajon surbodinance ', 'city under state jurisdiction in Latvia ', 'physiological condition ', 'Superior Conservatory of Music ', 'video albums discography ', 'art genre ', 'human who may be fictional ', 'medical profession ', 'historic house museum ', 'group of humans ', 'literary group ', 'municipality of Finland ', 'natural satellite ', 'crematorium ', 'eye disease ', 'Esperanto organization ', 'space agency ', 'consortium of universities in France ', 'theatrical troupe ', 'design school ', 'zoonosis ', 'ball game ', 'city municipality ', 'Japanese upper secondary school ', 'reed organ ', 'feature film ', 'sports league ', 'unincorporated community ', 'literature ', 'girl group ', 'Kreis in the kingdom of Bavaria ', 'television series ', 'type of business entity in the USA ', 'city of Pennsylvania ', 'plucked necked box lutes ', 'Grand Lodge ', 'hearing disorder ', 'religious movement ', 'viral video ', 'vassal state ', 'fraternity ', 'city in Cyprus ', 'federal office ', 'comune of Italy ', 'supercontinent ', 'airline ', 'castle chapel ', 'Oghuz languages ', 'law school ', 'television pilot ', 'activity ', 'Ortsteil ', 'moshav ', 'reservoir ', 'resistance movement ', 'music ', 'republic ', 'district of the canton of Schaffhausen ', 'municipality of Portugal ', 'geographical object ', 'metropolitan region in Germany ', 'Major Waterborne diseases ', 'television station ', 'genre ', 'Canadian football club ', 'Community of universities and higher education institutions ( France)', 'canton of Switzerland ', 'surgical procedure ', 'capital of a prefecture of Japan ', 'Abrahamic religion ', 'guitar technique ', 'provisional government ', 'military school ', 'musician ', 'political coalition ', 'murder ', 'athletics ', 'history of a country or state ', 'aspect of history ', 'municipality with town privileges in the Czech Republic ', 'religious occupation ', 'viols ', 'liberal arts college ', 'town divided by border ', 'private university ', 'original net animation ', 'Central Committee ', 'historic county of England ', 'natural sound ', 'original video animation ', 'Technische Hochschule ', 'Lutheran cathedral ', 'opera company ', 'borough of Pennsylvania ', 'security agency ', 'equestrian sport ', 'interior ministry ', 'public university ', 'crypt ', 'chef−lieu ', 'orchestra ', 'locality of Berlin ', 'science museum ', 'tapestry manufactory ', 'independent record label ', 'comic book series ', 'mediterranean sea ', 'Studium Generale ', 'supergroup ', 'character type ', 'former municipality of Switzerland ', 'village ', 'non−geologically related mountain range ', 'subregion ', 'sets of free reeds ', 'film ', 'non−fiction book ', 'national rugby union team ', 'province of China ', 'Constituent republics and provinces of Yugoslavia ', 'trade association ', 'ward of Japan ', 'position ', 'fraternal organization ', 'Broadcasting Board ', 'miniseries ', 'Wikimedia category ', 'property ', 'field of work ', 'specialty channel ', 'secondary school ', 'publisher ', 'slide trumpets ', 'record label ', 'short

film ', 'record company ', 'research ', 'urban municipality of Poland ', 'international organization ', 'consolidated city−county ', 'film series ', 'academic major ', 'sports venue ', 'American football team ', 'honor society ', 'liberal arts college in the United States ', 'state of Germany ', 'Category: Political parties in Russia ', 'character ', 'academy of sciences ', 'national university ', 'woodwind instrument ', 'Studentenverbindung ', 'empire ', 'boarding school ', 'group action ', 'open flutes with internal duct with fingerholes ', 'television series episode ', 'ch\^ateau ', 'transport company ', 'communist party ', 'Dichterkreis ', 'musical instrument ', 'archipelago ', 'necrosis ', 'municipality of Sweden ', 'concentration camp ', 'Gymnasium in Germany ', 'Empire on which the sun never sets ', 'academic profession ', 'district of the Czech Republic ', 'advocacy group ', 'county of Arizona ', 'political party in Catalonia ', 'special ward of Japan ', 'automobile marque ', 'non−controlled substance abuse ', 'region of Graub\"unden ', 'Indian Institutes of Technology ', 'consumer cooperative ', 'execution method ', 'municipal arrondissement ', 'prison ', 'market town ', 'printmaker ', 'video game publisher ', "employers' organization", 'core city of Japan ', 'public research university ', 'medical specialty ', 'traffic collision ', 'visual arts ', 'intergovernmental organization ', 'electronic musical instrument ', 'Public Scientific and Technical Research Establishment ', 'accident ', 'communication ', 'intelligence agency ', 'faculty of economics ', 'museum ', 'micronation ', 'baseball park ', 'covert operation ', 'legal professional ', 'broadcaster ', 'web documentary ', 'treatise ', 'international non−governmental organization ', 'municipality of Denmark ', 'administrative territorial entity of Poland ', 'archaeological site ', 'business ', 'subdistrict of the canton of Graub\"unden ', 'monotheistic religion ', 'meta−organization ', 'charter airline ', 'polytheism ', 'country ', 'visa policy ', 'county town ', 'punk band ', 'tourist attraction ', 'columbarium ', 'province of Prussia ', 'academic degree ', 'geographic location ', 'place with town rights and privileges ', 'capital ', 'lower house ', 'second−level administrative country subdivision ', 'musical profession ', 'greatest hits album ', 'international border ', 'human ', 'Regierungsbezirk ', 'institute of the Russian Academy of Sciences ', 'chemical compound ', 'area of engineering ', 'district of the canton of Solothurn ', 'village of New York ', 'disease ', 'state or insular area capital in the United States ', 'Christian denomination ', 'municipality of Puerto Rico ', 'architectural style ', 'boys school ', 'live video album ', 'association ', 'republic of the Soviet Union ', 'literary work ', 'city of regional significance of Ukraine ', 'collective pseudonym ', 'presidency of British India ', 'electronic organ ', 'music video compilation album ', 'branch of physics ', 'literary society ', 'enclave ', 'branch of chemistry ', 'park ', 'open−access publisher ', 'cultural region ', 'institute of technology ', 'commune of France ', 'land−grant university ', 'town of the United States ', 'mathematician ', 'collegiate university ', 'jurist ', 'research institute ', 'valley ', 'lyc\'ee ', 'system ', 'school of the French public service ', 'plucked string necked bowl lute ', 'television film ', 'humanistic gymnasium ', 'city of oblast significance ', 'artist ', 'term ', 'software ', 'plucked string instrument ', 'parish of the Church of Sweden ', 'music term ', 'music video ', 'barrio ', 'non−governmental organization ', 'provincial city ',

'independent school ', 'municipality of Catalonia ', 'continent
', 'magazine ', 'medical school ', "correspondents ' association
", 'urban area ', 'cemetery ', 'anime television series ', 'town
in Hungary ', 'Stadtbezirk ', 'corporate title ', 'animated
feature film ', 'Wikimedia list article ', 'university museum ',
'county of Connecticut ', 'superhero ', 'former municipality of
Sweden ', 'Inns of Court ', 'commune of Haiti ', 'television
program ', 'crime ', 'symptom ', 'neighborhood in Boston ', 'city
of the United States ', 'membranophones ', 'administration union
', 'electoral district of Finland ', 'abstract object ', 'state
in the Holy Roman Empire ', 'state of the United States ', '
subsidiary entity ', 'technology museum ', 'constituency of the
canton of St. Gallen ', 'genre of painting ', 'actor ', '
landlocked country ', 'higher school in the Empire of Japan ', '
upper−tier municipality ', 'filmmaking occupation ', 'type of
sport ', 'nation ', 'albums discography ', 'Jewish denomination ',
 'multi−purpose stadium ', 'limited liability company ', '
architectural firm ', 'municipalities and cities of Serbia ', '
gridiron football ', 'municipality seat ', 'league of towns ', '
ministry of communications ', 'province of Argentina ', '
national Church ', 'death ', 'government organization ', 'art
group ', 'continental area and surrounding islands ', 'treaty ',
'political faction ', 'audiovisual work ', 'specialty ', 'manga
series ', 'intoxication ', 'landform ', 'transcontinental country
', 'animated short film ', 'executive board ', 'drama school ', '
Eastern Orthodox patriarchate ', 'composition school ', 'team ',
'national sports team ', 'borough of New Jersey ', 'kommunaler
Spitzenverband ', 'state with limited recognition ', 'quarter of
 Hamburg ', 'guitar ', 'title of honor ', 'reredos ', 'homicide ',
'art school ', 'municipality of Liechtenstein ', 'regional
municipality of Ontario ', 'region of Belgium ', 'writers '
organization ', 'rail guided transport ', 'rural district of
Baden−W\"{u}rttemberg ', 'human action ', 'cultural movement ', '
institute ', 'sibling group ', 'poisoning ', 'national
association football team ', 'city ', 'private company ', '
literary form ', 'railway company ', 'municipality of Norway ', '
health professional ', 'state of Australia ', 'low−cost airline
', 'multi−sport club ', 'military academy ', 'association
football stadium ', 'narrative technique ', 'journalism school ',
 'educator ', 'saxophone ', 'state of India ', 'experience ', '
former liberal party ', 'learned society ', 'high island ', '
science fiction genre ', 'voluntary association ', 'university
building ', 'historical period ', 'high school ', 'county of New
Jersey ', 'Constitutional body ', 'century ', 'short story
collection ', 'city with millions of inhabitants ', 'research
university ', 'campus ', 'female idol group ', 'occurrence ', 'big
 city ', 'representation ', 'cover band ', 'work of art ', 'minor
basilica ', 'seminary ', 'group ', 'sports discipline ', 'occupied
 territory ', 'oblast of Russia ', 'governorate of the Russian
Empire ', 'county of Washington ', 'language regulator ', 'Place
of Execution ', 'political party in Spain ', 'veterans '
organization ', 'painting movement ', 'software company ', 'human
 settlement ', 'hospital ', 'organ ', 'internal bleeding ', '
municipality of Austria ', 'speculative fiction ', 'municipality
 of Spain ', 'county of Illinois ', 'Nazi concentration camp ', '
private school ', 'German public state broadcaster ', 'former
administrative territorial entity ', 'medical procedure ', 'web
series ', 'pressure group ', 'social movement ', 'true board
zithers with resonator box ', 'Landschaftsverband ', 'autonomous

country within the Kingdom of Denmark', 'village in the United States', 'Jewish cemetery', 'book publishing company', 'video game developer', 'metropolitan city of Italy', 'government region of North Rhine–Westphalia', 'district in Switzerland', 'foreign affairs ministry', 'prefecture–level city', 'prefecture of Japan', 'identifier', 'union territory of India', 'new religious movement', 'college athletic conference', 'town in Croatia', 'neighborhood', 'modern language', 'university in France', 'Green party', 'periodical', 'minister', 'single–tier municipality', 'clay animation film', 'former district of Switzerland', 'parliamentary term in the United Kingdom', 'juvenile political organisation', 'church building', 'secret police', 'auxiliary science', 'animated series', 'state of Austria', 'ethnic group', 'creative work', 'synthesizer', 'religious identity', 'television network', 'district of the canton of Schwyz', 'county of New York', 'college', 'rural district of Lower Saxony', 'philosophy', 'falling', 'mausoleum', 'ocean', 'Ortschaft', 'dependent territory', 'race', 'province of Canada', 'political party in Croatia', 'rural district of Rhineland–Palatinate', 'Belgian municipality with city privileges', 'dialect', 'social science', 'government', 'rural cemetery', 'military museum', 'musical work', 'county of California', 'Catholic university', 'book series', 'volleyball team', 'superior graduate school in Italy', 'singing duo', 'percussion instrument', 'engineer', 'rugby union team', 'graduate school', 'mockumentary', "Conceyu d'Asturies", 'orthodox cathedral', 'VIA', 'trade union', 'public service', 'sorority', 'former French region', 'historical society', 'East Slavic languages', 'group of interconnected lakes', 'musical duo', 'borough', 'quarter', 'registered association', 'constituent part of the United Kingdom', 'theatrical genre', 'legislative assembly', 'area of law', 'songwriter', 'public educational institution of the United States', 'university of applied sciences', 'former municipality', 'metropolitan area', 'Japanese television drama', 'League of Nations mandate', 'Federal Ministry in Germany', 'nonprofit organization', 'online newspaper', 'cause of death', 'private not–for–profit educational institution', 'municipiu of Romania', 'county–level city', "director's cut", 'city with powiat rights', 'administrative territorial entity', 'daily newspaper', 'tapestry series', 'rockumentary', 'higher party school', 'association football league', 'state school', 'heavy metal band', 'borough of New York City', 'military operation', 'school', 'evaluation', 'gymnasium', 'protectorate', 'drinking fountain', 'chemical hazard', 'comic group', 'cultural institution', 'political parties in Germany', 'tag team', 'book', 'college of music', 'psychology', 'juridical person', 'foundation', 'historic county of the United Kingdom', 'mallet percussion instrument', 'military personnel', 'parliamentary group', 'accordion', 'Relajaci\'on', 'international court', 'island', 'constituency of the canton of Lucerne', 'Esperanto language institute', 'academic discipline', 'fictional human', 'steering committee', 'madhhab', 'parliament', 'governorate', 'federally funded research and development center', 'arrondissement of France', 'massif', 'live–action animated film', 'anglican or episcopal cathedral', 'art form', 'home rule municipality of Pennsylvania', 'upper house', 'comedy', 'biology', 'city of Japan', 'village of New Jersey', 'international airport', 'political ideology', '

principality ', 'democratic republic ', 'acoustic guitar ', '
salon ', 'New England town ', 'sea ', 'county of Minnesota ', '
history ', 'arts educational institution ', 'city designated by
government ordinance ', 'cultural heritage site in Russia ', '
military officer ', 'statutory city of Austria ', 'constituent
state ', 'public broadcasting ', 'credit institution ', '
transport accident ', 'head and neck disease ', 'military unit
branch−size class ', 'central bank ', 'music genre ', '
territorial entity ', 'television channel ', 'literary genre ', '
musical group ', 'colony ', 'Landeskirche ', 'concept ', 'branch
of science ', 'natural language ', 'federative unit of Brazil ',
'play ', 'oblast of Ukraine ', 'advisory board ', 'borough of
Hamburg ', 'single−reed instrument ', 'music scene ', 'divided
country ', 'science fiction ', 'valve horn ', 'civil parish ', '
musical technique ', 'major label ', 'Oriental studies ', '
hardback ', 'written work ', 'international financial
institution ', 'atheneum ', 'video game theme ', 'region of
Finland ', 'ecclesiastical title ', 'constituency ', 'supreme
court ', 'district of the canton of Valais ', 'Japanese TV
series ', 'engineering school ', 'city or town ', 'joint−stock
company ', 'area of London ', 'home rule city of Michigan ', '
film studio ', 'English ', 'science ', 'former provinces of Italy
', 'Landeswohlfahrtsverband ', 'labour party ', 'secret society
', 'district of Israel ', 'first−level administrative country
subdivision ', 'professional sports league ', 'Khanate ', 'state
', 'United States federal executive department ', 'mountain ', '
journalism genre ', 'region of Italy ', 'religious text ', '
district of the canton of Neuch\^atel ', 'town ', 'Catholic
religious occupation ', 'municipal arrondissement of Marseille
', 'superpower ', 'memory institution ', 'religious denomination
', 'academic institution ', 'people ', 'artist collective ', '
holding company ', 'municipality of Switzerland ', 'serial film
', 'philosophical movement ', 'urban area in Sweden ', '
scientific society ', 'Hanseatic city ', 'social influence ', '
district of the canton of Fribourg ', 'imperial university of
the Russian Empire ', 'state university system ', 'pontifical
university ', 'religious organization ', 'state church ', '
overseas department of France ', 'ceremonial county of England
', 'shock troops ', 'county of Maryland ', 'radio station ', '
tradesperson ', 'Metropolitan Statistical Area ', 'facility ', '
organization related to nonviolence ', 'public scientific,
cultural or professional establishment ', 'charter city ', '
class of instruments ', 'college of the University of Oxford ',
'province of the Republic of China ', 'district of the canton
of Ticino ', 'film genre ', 'believer ', 'aviation accident ', '
keyboard ', 'mountain range ', 'health problem ', 'anime and
manga genre ', 'volcano ', 'Ecole secondaire de Neuchatel ', '
device ', 'drama ', 'dead language ', 'statistical service ', '
physics ', 'commercial organization ', 'keyboard instrument ', '
parish of Jamaica ', 'historical language ', 'literary movement
', 'culture ', 'business school ', 'cinematic technique ', '
enterprise ', 'video game genre ', 'atypical pneumonia ', '
faculty ', 'art music ', 'district of the canton of Vaud ', '
necked bowl lutes sounded by plectrum ', 'rural district of
North Rhine−Westphalia ', 'age ', 'United States national
cemetery ', 'single oboes with conical bore ', 'philosophical
school ', "UCI Women's Team", 'educational institution ', '
metropolitan municipality of South Africa ', 'Bundestag
committee ', 'basketball team ', 'supranational union ', '

apostasy', 'ethics', 'economic branch', 'grande \'{e}cole', 'administrative district of the canton of Bern', 'urban municipality of Germany', 'census−designated place', 'largest city', 'theatre', 'sovereign state', 'international school', 'municipality', 'district or neighborhood of Los Angeles', 'public policy school', 'academia', 'geographic region', 'association football club', 'extermination camp', 'profession', 'terrorist organization', 'symphonic orchestra', 'airport', 'cathedral', 'computing platform', 'locality of Mexico', 'college of the University of Cambridge', 'liberal religion', 'government region of Baden–W\"{u}rttemberg', 'medical society', 'specialized agency of the United Nations', 'language', 'city/town', 'community center', 'proposed country', 'historical Chinese state', 'faculty of law', 'spoken language', 'zone of Nepal', 'protection', 'independent city of Germany', 'higher education institution', 'stop−motion animated film', 'fictional city', 'German Student Corps', 'voivodeship of Poland', 'baseball team', 'department of Argentina', 'district of the canton of Aargau', 'UCI Trade Team II', 'county of Virginia', 'news agency', 'newspaper', 'athletic conference', 'historical country', 'rural district of Saxony', 'politician', 'border town', 'infectious disease', ''district of the canton of Z\"{u}rich'', 'mind sport', 'pop duo', 'Swedish Royal Academies', 'world view', 'Ausbildungsberuf', 'staff college', 'neighborhood in Brooklyn, New York City', 'Eastern Catholic Church', 'department of France', 'city in New Jersey', 'political party', 'primary school', 'boy band', 'art museum', 'university', 'animated character', 'municipality of Belgium', 'type of business entity', 'organization', 'conservatory', 'municipality of Vietnam', 'entertainment company', 'London borough', 'county of Massachusetts', 'instrumentalist', 'city of New Brunswick', 'historical region', 'group of fictional characters', 'association football team', 'British Overseas Territories', 'absence', 'confederation', 'subdistrict of the canton of Ticino', 'Crusader states', 'city−state', 'Catholic cathedral', 'jurisdiction', 'municipality of the Czech Republic', 'Protestantism', 'education', 'sports club', 'university college', 'special city of Japan', 'subcontinent', 'royal palace', 'writing circle', 'clinical sign', 'clan', '3D film', 'film production company', 'human−geographic territorial entity', 'sequel film', 'building', 'colonial power', 'oboe family instrument', 'area of mathematics', 'vascular disease', 'academy', 'island of Japan', 'fountain'].

Please extract a list of all the entities of the types described above from the given passage.

Please provide your response in valid JSON using the following response schema: {'type': 'object', 'properties': {'entities': {'type': 'array', 'items': {'type': 'string'}}}, 'required': ['entities'], 'additionalProperties': False, 'strict': True}

For example:

Example input: "Renate Axt was a writer, born in Darmstadt in Germany. Germany shares a border with Austria and Czechoslovakia."

```
Expected response: {{
    entities: [
        "Renate Axt",
        "Darmstadt",
        "Germany",
        "Austria",
        "Czechoslovakia",
    ]
}}
```

**Codex - KG Extraction Prompt Template**

You will be provided with a text describing the directed relationships between various entities

You will also be provided with a list of entities contained within that statement.

Entities are related to the other entities via the following directed relationships: ['country of citizenship', 'country', 'occupation', 'place of birth', 'member of political party', 'educated at', 'genre', 'member of', 'located in the administrative terroritorial entity', 'languages spoken, written, or signed', 'religion', 'instrument', 'sibling', 'place of death', 'shares border with', 'spouse', 'place of burial', 'cast member', 'record label', 'field of work', 'employer', 'influenced by', 'location of formation', 'diplomatic relation', 'cause of death', 'country of origin', 'residence', 'airline hub', 'official language', 'narrative location', 'capital', 'ethnic group', 'member of sports team', 'language of work or name', 'time period', 'headquarters location', 'child', 'sport', 'medical condition', 'movement', 'director', 'uses', 'founded by', 'parent organization', 'continent', 'occupant', 'mountain range', 'symptoms', 'part of', 'publisher', 'drug used for treatment', 'industry', 'named after', 'unmarried partner', 'airline alliance', 'creator', 'legal form', 'author', 'chairperson', 'health specialty', 'architect', 'chief executive officer', 'product or material produced', 'architectural style', 'legislative body', 'practiced by', 'foundational text', 'studies', 'use']

Your goal is to extract the relationships between the entities as a directed graph and represent that graph as a list of triples in valid json using the following schema: {'type': 'object', 'properties': {'triples': {'type': 'array', 'items': {'type': 'object', 'properties': {'source_node': {'type': 'object', 'properties': {'name': {'type': 'string'}}, 'required': ['name'], 'additionalProperties': False}, 'relation': {'type': 'object', 'properties': {'name': {'type': 'string'}}, 'required': ['name'], 'additionalProperties': False}, 'target_node': {'type': 'object', 'properties': {'name': {'type': 'string'}}, 'required': ['name'], 'additionalProperties': False}}, 'required': ['source_node', 'relation', 'target_node'], 'additionalProperties': False}}}, 'required': ['triples'], 'additionalProperties': False, 'strict': True}

The triples should be represented in directed order with the relation direction going from "source_node" to "target_node"

Examples:

1. Example entities: ["Renate Axt", "Darmstadt", "Germany", "Austria", "Czechoslovakia"]
1. Example text: "Renate Axt was a writer, born in Darmstadt in Germany. Germany shares a border with Austria and Czechoslovakia."

1. Expected output: {{
   "triples": [
       {{'source_node': {{'name': 'Renate Axt'}}, 'relation': {{'name': 'occupation'}}, 'target_node': {{'name': 'writer'}}}},
       {{'source_node': {{'name': 'Renate Axt'}}, 'relation': {{'name': 'place of birth'}}, 'target_node': {{'name': 'Darmstadt'}}}},
       {{'source_node': {{'name': 'Darmstadt'}}, 'relation': {{'name': 'country'}}, 'target_node': {{'name': 'Germany'}}}}
       {{'source_node': {{'name': 'Germany'}}, 'relation': {{'name': 'shares border with'}}, 'target_node': {{'name': 'Austria'}}}},
       {{'source_node': {{'name': 'Germany'}}, 'relation': {{'name': 'shares border with'}}, 'target_node': {{'name': 'Czechoslovakia'}}}},
   ]
}}

2. Example entities: ["Rome", "Roman Republic", "ancient Rome", "Western Roman Empire", "Persian Empire", "classical antiquity", "Latin"]
2. Example text: "Rome has been located in a range of administrative terroritorial entities including the Roman Republic, ancient Rome and the Western Roman Empire. Ancient Rome, which shared a border with the Perian Empire, existed in the time period of classical antiquity. Within the Western Roman Empire, the official language was Latin."

2. Expected output: {{
   "triples": [
       {{'source_node': {{'name': 'Rome'}}, 'relation': {{'name': 'located in the administrative terroritorial entity'}}, 'target_node': {{'name': 'Roman Republic'}}}},
       {{'source_node': {{'name': 'Rome'}}, 'relation': {{'name': 'located in the administrative terroritorial entity'}}, 'target_node': {{'name': 'ancient Rome'}}}},
       {{'source_node': {{'name': 'Rome'}}, 'relation': {{'name': 'located in the administrative terroritorial entity'}}, 'target_node': {{'name': 'Western Roman Empire'}}}},
       {{'source_node': {{'name': 'ancient Rome'}}, 'relation': {{'name': 'shares border with'}}, 'target_node': {{'name': 'Persian Empire'}}}},
       {{'source_node': {{'name': 'ancient Rome'}}, 'relation': {{'name': 'time period'}}, 'target_node': {{'name': 'classical antiquity'}}}},

```
        {{'source_node': {{'name': 'Western Roman Empire'}}, '
            relation': {{'name': 'official language'}}, '
            target_node': {{'name': 'Latin'}}}},
    ]
}}
```

3. Example entities: ["G.I. Joe: The Rise of Cobra", "Lee Byung-hun", "Marlon Wayans", "film director", "New York City", "Howard University", "film actor", "Seoul", "Korean"]

3. Example text: "G.I. Joe: The Rise of Cobra, a film that takes place in Tokyo, starred Lee Byung-hun and Marlon Wayans. Wayans, also a film-director was born in New York City and attended Howard University. Meanwhile Lee Byung-hun is an actor born in Seoul who speaks Korean."

```
[
    {{'source_node': {{'name': 'G.I. Joe: The Rise of Cobra'}}, '
        relation': {{'name': 'cast member'}}, 'target_node': {{'
        name': 'Lee Byung-hun'}}}},
    {{'source_node': {{'name': 'G.I. Joe: The Rise of Cobra'}}, '
        relation': {{'name': 'cast member'}}, 'target_node': {{'
        name': 'Marlon Wayans'}}}},
    {{'source_node': {{'name': 'G.I. Joe: The Rise of Cobra'}}, '
        relation': {{'name': 'narrative location'}}, 'target_node
        ': {{'name': 'Tokyo'}}}},
    {{'source_node': {{'name': 'Marlon Wayans'}}, 'relation': {{'
        name': 'occupation'}}, 'target_node': {{'name': 'film
        director'}}}},
    {{'source_node': {{'name': 'Marlon Wayans'}}, 'relation': {{'
        name': 'place of birth'}}, 'target_node': {{'name': 'New
        York City'}}}},
    {{'source_node': {{'name': 'Marlon Wayans'}}, 'relation': {{'
        name': 'educated at'}}, 'target_node': {{'name': 'Howard
        University'}}}},
    {{'source_node': {{'name': 'Lee Byung-hun'}}, 'relation': {{'
        name': 'occupation'}}, 'target_node': {{'name': 'film
        actor'}}}},
    {{'source_node': {{'name': 'Lee Byung-hun'}}, 'relation': {{'
        name': 'place of birth'}}, 'target_node': {{'name': 'Seoul
        '}}}},
    {{'source_node': {{'name': 'Lee Byung-hun'}}, 'relation': {{'
        name': 'languages spoken, written, or signed'}}, '
        target_node': {{'name': 'Korean'}}}},
]
```

**FinDKG - Entity Extraction Prompt Template**

You will be provided with a statement describing the relationship between various entities.

The different entities described are of the following types: ['ORG/GOV', 'ORG', 'PERSON', 'SECTOR', 'ORG/REG', 'EVENT', 'ECON_INDICATOR', 'FIN_INSTRUMENT', 'COMP', 'GPE', 'CONCEPT', 'PRODUCT'].

Please extract a list of all the entities of the types described above from the given passage.

Please provide your response in valid JSON using the following
    response schema: {'type': 'object', 'properties': {'entities':
    {'type': 'array', 'items': {'type': 'string'}}}, 'required':
    ['entities'], 'additionalProperties': False, 'strict': True}

For example:

Example input: "Italian debt is related to policies of the
    European Central Bank. This debt has an impact on the Italian
    Government in addition to investors. Investors may also
    subsequently impact the value of Yuan."

Expected response: {{
    "entities": [
        "Italian debt",
        "European Central Bank",
        "Italian Government",
        "investors",
        "Yuan",
    ]
}}

**FinDKG - KG Extraction Prompt Template**

You will be provided with a text describing the directed
    relationships between various entities

You will also be provided with a list of entities contained within
    that statement.

Entities are related to the other entities via the following
    directed relationships: ['Control', 'Impact', 'Participates_In
    ', 'Relate_To', 'Operate_In', 'Positive_Impact_On', 'Raise', '
    Announce', 'Introduce', 'Negative_Impact_On', 'Is_Member_Of',
    'Decrease', 'Has', 'Produce', 'Invests_In']"

Relation Definitions:
– Has: Indicates ownership or possession, often of assets or
    subsidiaries in a financial context.
– Announce: Refers to the formal public declaration of a financial
    event, product launch, or strategic move.
– Operate_In: Describes the geographical market in which a
    business entity conducts its operations.
– Introduce: Denotes the first–time introduction of a financial
    instrument, product, or policy to the market.
– Produce: Specifies the entity responsible for creating a
    particular product, often in a manufacturing or financial
    product context.
– Control: Implies authority or regulatory power over monetary
    policy, financial instruments, or market conditions.
– Participates_In: Indicates active involvement in an event that
    has financial or economic implications.
– Impact: Signifies a notable effect, either positive or negative,
    on market trends, financial conditions, or economic
    indicators.
– Positive_Impact_On: Highlights a beneficial effect on financial
    markets, economic indicators, or business performance.
– Negative_Impact_On: Underlines a detrimental effect on financial
    markets, economic indicators, or business performance.

- Relate_To: Points out a connection or correlation with a financial concept, sector, or market trend.
- Is_Member_Of: Denotes membership in a trade group, economic union, or financial consortium.
- Invests_In: Specifies an allocation of capital into a financial instrument, sector, or business entity.
- Raise: Indicates an increase, often referring to capital, interest rates, or production levels in a financial context.
- Decrease: Indicates a reduction, often referring to capital, interest rates, or production levels in a financial context.

Your goal is to extract the relationships between the entities as a directed graph and represent that graph as a list of triples in valid json using the following schema: {'type': 'object', 'properties': {'triples': {'type': 'array', 'items': {'type': 'object', 'properties': {'source_node': {'type': 'object', 'properties': {'name': {'type': 'string'}}, 'required': ['name'], 'additionalProperties': False}, 'relation': {'type': 'object', 'properties': {'name': {'type': 'string'}}, 'required': ['name'], 'additionalProperties': False}, 'target_node': {'type': 'object', 'properties': {'name': {'type': 'string'}}, 'required': ['name'], 'additionalProperties': False}}, 'required': ['source_node', 'relation', 'target_node'], 'additionalProperties': False}}}, 'required': ['triples'], 'additionalProperties': False, 'strict': True}

The triples should be represented in directed order with the relation direction going from "source_node" to "target_node"

Examples:

1. Example entities = ["U.S. Air Force", "Asia and Europe", "Afghanistan", "government"]
1. Example text = "The U.S. Air Force controls Asia and Europe, and operates in Afghanistan. Afghanistan has a government."

1. Expected output: {{
   "triples": [
       {{'source_node': {{'name': 'U.S. Air Force'}}, 'relation': {{'name': 'Control'}}, 'target_node': {{'name': 'Asia and Europe'}}}},
       {{'source_node': {{'name': 'U.S. Air Force'}}, 'relation': {{'name': 'Operate_In'}}, 'target_node': {{'name': 'Afghanistan'}}}},
       {{'source_node': {{'name': 'Afghanistan'}}, 'relation': {{'name': 'Has'}}, 'target_node': {{'name': 'government'}}}},
   ]
}}

2. Example entities = ["Tax Cut", "consumer spending", "investment", "Economic indicators", "The U.S. Economy"]
2. Example text = "Tax cuts can impact investment but also have a positive impact on consumer spending which relates to important economic indicators and can impact the U.S. economy."

```
2. Expected output: {{
   "triples": [
       {{'source_node': {{'name': 'Tax Cut'}}, 'relation': {{'
           name': 'Positive_Impact_On'}}, 'target_node': {{'name
           ': 'Consumer Spending'}}}},
       {{'source_node': {{'name': 'Tax Cut'}}, 'relation': {{'
           name': 'Impact'}}, 'target_node': {{'name': '
           investment'}}}},
       {{'source_node': {{'name': 'Consumer Spending'}}, '
           relation': {{'name': 'Relate_To'}}, 'target_node': {{'
           name': 'Economic indicators'}}}},
       {{'source_node': {{'name': 'Consumer Spending'}}, '
           relation': {{'name': 'Impact'}}, 'target_node': {{'
           name': 'The U.S. Economy'}}}},
   ]
}}

3. Example entities: ["Federal Reserve System", "Gold", "Expenses
   ", "The U.S. Economy", "Gross Domestic Product", "U.S. stocks
   "]
3. Example text = "The Federal Reserve System controls expenses,
   which can have a positive impact on Gold, an asset also
   impacted by the Federal Reserve System. Additionally this
   system controls the U.S. Economy which has a relationship with
    Gross Domestic Product and U.S. stocks."

3. Expected output: {{
   "triples": [
       {{'source_node': {{'name': 'Federal Reserve System'}}, '
           relation': {{'name': 'Impact'}}, 'target_node': {{'
           name': 'Gold'}}}},
       {{'source_node': {{'name': 'Federal Reserve System'}}, '
           relation': {{'name': 'Control'}}, 'target_node': {{'
           name': 'Expenses'}}}},
       {{'source_node': {{'name': 'Federal Reserve System'}}, '
           relation': {{'name': 'Control'}}, 'target_node': {{'
           name': 'The U.S. Economy'}}}},
       {{'source_node': {{'name': 'The U.S. Economy'}}, 'relation
           ': {{'name': 'Relate_To'}}, 'target_node': {{'name': '
           Gross Domestic Product'}}}},
       {{'source_node': {{'name': 'The U.S. Economy'}}, 'relation
           ': {{'name': 'Relate_To'}}, 'target_node': {{'name': '
           U.S. stocks'}}}},
       {{'source_node': {{'name': 'Expenses'}}, 'relation': {{'
           name': 'Positive_Impact_On'}}, 'target_node': {{'name
           ': 'Gold'}}}},
   ]
]
```

**Globi - Entity Extraction Prompt Template**

Due to the prohibitively large number of edges in the Globi KG, the prompt template edge-list is truncated here to fit in the manuscript. The prompt template used in validation contained all available edges.

```
You will be provided with a statement describing the relationship
    between various entities.

The different entities described are of the following types: ['
    Cicindellidae', 'Euphausiidae', ..., 'Centracanthidae'].
```

Please extract a list of all the entities of the types described above from the given passage.

Please provide your response in valid JSON using the following response schema: {'type': 'object', 'properties': {'entities': {'type': 'array', 'items': {'type': 'string'}}}, 'required': ['entities'], 'additionalProperties': False, 'strict': True}

For example:

Example input: "Chromatomyia erigerontophaga is a species known to visit the flowers of Potentilla nivea which in turn have interactions with Salix arctica and Draba nivalis. C. erigerontophaga also pollinate Erigeron compositus. This flower is known to interact with Populus tremuloides and Luetkea pectinata."

Expected response: {{
    "entities": [
        "Chromatomyia erigerontophaga",
        "Potentilla nivea",
        "Salix arctica",
        "Draba nivalis",
        "Erigeron compositus",
        "Populus tremuloides",
        "Luetkea pectinata",
    ]
}}

**Globi - KG Extraction Prompt Template**

You will be provided with a text describing the directed relationships between various entities

You will also be provided with a list of entities contained within that statement.

Entities are related to the other entities via the following directed relationships: ['parasiteOf', 'hasHost', 'eats', 'preysOn', 'pollinates', 'pathogenOf', 'visitsFlowersOf', 'hasVector', 'rootparasiteOf', 'endoparasiteOf', 'interactsWith', 'kills', 'createsHabitatFor', 'parasitoidOf', 'hasRoost', 'coRoostsWith', 'ecologicallyRelatedTo', 'epiphyteOf', 'commensalistOf', 'mutualistOf', 'providesNutrientsFor', 'ectoparasiteOf', 'coOccursWith', 'hasHabitat', 'symbiontOf', 'kleptoparasiteOf', 'adjacentTo', 'allelopathOf', 'laysEggsIn', 'visits', 'hyperparasiteOf', 'laysEggsOn', 'endoparasitoidOf', 'livesOn', 'guestOf', 'livesInsideOf', 'ectoParasitoid', 'livesNear', 'livesUnder', 'inhabits', 'hasDispersalVector']

Your goal is to extract the relationships between the entities as a directed graph and represent that graph as a list of triples in valid json using the following schema: {'type': 'object', 'properties': {'triples': {'type': 'array', 'items': {'type': 'object', 'properties': {'source_node': {'type': 'object', 'properties': {'name': {'type': 'string'}}, 'required': ['name'], 'additionalProperties': False}, 'relation': {'type': 'object', 'properties': {'name': {'type': 'string'}}, 'required': ['name'], 'additionalProperties': False}, 'target_node': {'

```
type': 'object', 'properties': {'name': {'type': 'string'}}, '
    required': ['name'], 'additionalProperties': False}}, '
    required': ['source_node', 'relation', 'target_node'], '
    additionalProperties': False}}}, 'required': ['triples'], '
    additionalProperties': False, 'strict': True}
```

The triples should be represented in directed order with the
    relation direction going from "source_node" to "target_node"

Examples:

1. Example entities: ["Chromatomyia erigerontophaga", "Potentilla
    nivea", "Salix arctica", "Draba nivalis", "Erigeron compositus
    ", "Populus tremuloides", "Luetkea pectinata"]
1. Example text: "Chromatomyia erigerontophaga is a species known
    to visit the flowers of Potentilla nivea which in turn have
    interactions with Salix arctica and Draba nivalis. C.
    erigerontophaga also pollinate Erigeron compositus. This
    flower is known to interact with Populus tremuloides and
    Luetkea pectinata."

1. Expected output: {{
    "triples": [
        {{'source_node': {{'name': 'Chromatomyia erigerontophaga
            '}}, 'relation': {{'name': 'visitsFlowersOf'}}, '
            target_node': {{'name': 'Potentilla nivea'}}}},
        {{'source_node': {{'name': 'Chromatomyia erigerontophaga
            '}}, 'relation': {{'name': 'pollinates'}}, '
            target_node': {{'name': 'Erigeron compositus'}}}},
        {{'source_node': {{'name': 'Potentilla nivea'}}, 'relation
            ': {{'name': 'interactsWith'}}, 'target_node': {{'name
            ': 'Salix arctica'}}}},
        {{'source_node': {{'name': 'Potentilla nivea'}}, 'relation
            ': {{'name': 'interactsWith'}}, 'target_node': {{'name
            ': 'Draba nivalis'}}}},
        {{'source_node': {{'name': 'Erigeron compositus'}}, '
            relation': {{'name': 'interactsWith'}}, 'target_node':
             {{'name': 'Populus tremuloides'}}}},
        {{'source_node': {{'name': 'Erigeron compositus'}}, '
            relation': {{'name': 'interactsWith'}}, 'target_node':
             {{'name': 'Luetkea pectinata'}}}}
    ]
}}

2. Example entities: ["Pinus jeffreyi", "Betula occidentalis", "
    Wyethia mollis", "Collomia heterophylla"]
2. Example text: "Pinus jeffreyi interacts with several species
    including Betula occidentalis and Wyethia mollis. Wyethia
    mollis in turn interacts with Collomia heterophylla."

2. Expected output: {{
    "triples": [
        {{'source_node': {{'name': 'Pinus jeffreyi'}}, 'relation':
             {{'name': 'interactsWith'}}, 'target_node': {{'name':
             'Betula occidentalis'}}}},
        {{'source_node': {{'name': 'Pinus jeffreyi'}}, 'relation':
             {{'name': 'interactsWith'}}, 'target_node': {{'name':
             'Wyethia mollis'}}}},
```

```
            {{'source_node': {{'name': 'Wyethia mollis'}}, 'relation':
                {{'name': 'interactsWith'}}, 'target_node': {{'name':
                'Collomia heterophylla'}}}}}}
    ]
}}
```

3. Example entities: ["Neotoma mexicana", "Lynx rufus", "Canis latrans", "Prunus serotina", "Sylvilagus cunicularius", "Panthera leo", "Crotalus pricei", "Sceloporus jarrovii", "Junco phaeonotus"]

3. Example text: "Crotalus pricei is a species known to prey on several species including, Sceloporus jarrovii, Junco phaeonotus, and Neotoma mexicana. Neotoma mexicana have interactions with several species including Lynx rufus and Canis latrans. Canis latrans are known to co-occur with Panthera leo and eat Prunus serotina and Sylvilagus cunicularius."

3. Expected output:
```
{{
  "triples": [
      {{'source_node': {{'name': 'Neotoma mexicana'}}, 'relation
          ': {{'name': 'interactsWith'}}, 'target_node': {{'name
          ': 'Lynx rufus'}}}}}},
      {{'source_node': {{'name': 'Neotoma mexicana'}}, 'relation
          ': {{'name': 'interactsWith'}}, 'target_node': {{'name
          ': 'Canis latrans'}}}}}},
      {{'source_node': {{'name': 'Canis latrans'}}, 'relation':
          {{'name': 'eats'}}, 'target_node': {{'name': 'Prunus
          serotina'}}}}}},
      {{'source_node': {{'name': 'Canis latrans'}}, 'relation':
          {{'name': 'eats'}}, 'target_node': {{'name': '
          Sylvilagus cunicularius'}}}}}},
      {{'source_node': {{'name': 'Canis latrans'}}, 'relation':
          {{'name': 'coOccursWith'}}, 'target_node': {{'name': '
          Panthera leo'}}}}}},
      {{'source_node': {{'name': 'Crotalus pricei'}}, 'relation
          ': {{'name': 'preysOn'}}, 'target_node': {{'name': '
          Sceloporus jarrovii'}}}}}},
      {{'source_node': {{'name': 'Crotalus pricei'}}, 'relation
          ': {{'name': 'preysOn'}}, 'target_node': {{'name': '
          Neotoma mexicana'}}}}}},
      {{'source_node': {{'name': 'Crotalus pricei'}}, 'relation
          ': {{'name': 'preysOn'}}, 'target_node': {{'name': '
          Junco phaeonotus'}}}}}},
  ]
}}
```

**Oregano - Entity Extraction Prompt Template**

You will be provided with a statement describing the relationship between various entities.

The different entities described are of the following types: ['COMPOUND', 'GENE', 'DISEASE', 'PROTEIN', 'MOLECULE', 'ACTIVITY', 'EFFECT', 'PHENOTYPE', 'PATHWAY', 'INDICATION', 'SIDE_EFFECT'].

Please extract a list of all the entities of the types described above from the given passage.

Please provide your response in valid JSON using the following
    response schema: {'type': 'object', 'properties': {'entities':
    {'type': 'array', 'items': {'type': 'string'}}}, 'required':
    ['entities'], 'additionalProperties': False, 'strict': True}

For example:

Example input: "The compound kizuta saponin k12 targets
    prostaglandin G/H synthase 2. The prostaglandin G/H synthase 2
    is a gene product of PTGS2, which acts within the pathway of
    the synthesis of 15−eicosatetraenoic acid derivatives."

Expected response: {{
    "entities": [
        "kizuta saponin k12",
        "prostaglandin G/H synthase 2",
        "PTGS2",
        "synthesis of 15−eicosatetraenoic acid derivatives",
    ]
}}

**Oregano - KG Extraction Prompt Template**

You will be provided with a text describing the directed
    relationships between various entities

You will also be provided with a list of entities contained within
    that statement.

Entities are related to the other entities via the following
    directed relationships: ['has_target', 'increase_activity', '
    has_activity', 'decrease_activity', 'increase_effect', '
    has_effect', 'decrease_effect', 'increase_efficacy', '
    decrease_efficacy', 'causes_condition', 'has_phenotype', '
    is_affecting', 'is_substance_that_treats', 'acts_within', '
    has_indication', 'has_side_effect', 'gene_product_of']

Your goal is to extract the relationships between the entities as
    a directed graph and represent that graph as a list of triples
    in valid json using the following schema: {'type': 'object',
    'properties': {'triples': {'type': 'array', 'items': {'type':
    'object', 'properties': {'source_node': {'type': 'object', '
    properties': {'name': {'type': 'string'}}, 'required': ['name
    '], 'additionalProperties': False}, 'relation': {'type': '
    object', 'properties': {'name': {'type': 'string'}}, 'required
    ': ['name'], 'additionalProperties': False}, 'target_node': {'
    type': 'object', 'properties': {'name': {'type': 'string'}}, '
    required': ['name'], 'additionalProperties': False}}, '
    required': ['source_node', 'relation', 'target_node'], '
    additionalProperties': False}}}, 'required': ['triples'], '
    additionalProperties': False, 'strict': True}

The triples should be represented in directed order with the
    relation direction going from "source_node" to "target_node"

The valid edge directions are ['COMPOUND −> PROTEIN', 'COMPOUND −>
    MOLECULE', 'COMPOUND −> ACTIVITY', 'COMPOUND −> EFFECT', '
    COMPOUND −> COMPOUND', 'GENE −> DISEASE', 'DISEASE −>
    PHENOTYPE', 'COMPOUND −> GENE', 'COMPOUND −> DISEASE', 'GENE

-> PATHWAY', 'COMPOUND -> INDICATION', 'COMPOUND -> SIDE', '
PROTEIN -> GENE']

Examples:

1. Example entities: ["kizuta saponin k12", "prostaglandin G/H
   synthase 2", "PTGS2", "synthesis of 15-eicosatetraenoic acid
   derivatives"]
1. Example text: "The compound kizuta saponin k12 targets
   prostaglandin G/H synthase 2. The prostaglandin G/H synthase 2
   is a gene product of PTGS2, which acts within the pathway of
   the synthesis of 15-eicosatetraenoic acid derivatives."

1. Expected output: {{
   "triples": [
       {{"source_node": {{"name": "kizuta saponin k12"}}, "
          relation": {{"name": "has_target"}}, "target_node":
          {{"name": "prostaglandin G/H synthase 2"}}}},
       {{"source_node": {{"name": "prostaglandin G/H synthase
          2"}}, "relation": {{"name": "gene_product_of"}}, "
          target_node": {{"name": "PTGS2"}}}},
       {{"source_node": {{"name": "PTGS2"}}, "relation": {{"name
          ": "acts_within"}}, "target_node": {{"name": "
          synthesis of 15-eicosatetraenoic acid derivatives
          "}}}},
   ]
}}

2. Example entities: ["xk469", "aldehyde oxidase 1", "toxic liver
   disease", "neoplasms"]
2. Example text: "The drug xk469 has an effect on aldehyde oxidase
   1. Interestingly, alterations in aldehyde oxidase 1 are known
   to cause several conditions including toxic liver disease,
   and neoplasms."

2. Expected output: {{
   "triples": [
       {{'source_node': {{'name': 'xk469'}}, 'relation': {{'name
          ': 'is_affecting'}}, 'target_node': {{'name': '
          aldehyde oxidase 1'}}}}]"
       {{'source_node': {{'name': 'aldehyde oxidase 1'}}, '
          relation': {{'name': 'causes_condition'}}, '
          target_node': {{'name': 'toxic liver disease'}}}},
       {{'source_node': {{'name': 'aldehyde oxidase 1'}}, '
          relation': {{'name': 'causes_condition'}}, '
          target_node': {{'name': 'neoplasms'}}}},
   ]
}}

3. Example entities: ["Aniracetam", "Dopamine D2 receptor", "DRD2
   ", "Magnesium Sulfate", "Paramethadione", "Dihydrocodeine", "
   Orvepitant"]
3. Example text: "'The drug known as Aniracetam targets the
   Dopamine D2 receptor, a gene product of DRD2. Aniracetam is
   known to enhance the efficacy of Magnesium Sulfate, which in
   turn boosts the effects of Paramethadione, Dihydrocodeine, and

Orvepitant. Therefore, caution should be taken when using Aniracetam due to its wide-reaching effects.

3. Expected output: {{
    "triples": [
        {{'source_node': {{'name': 'Aniracetam'}}, 'relation': {{'name': 'has_target'}}, 'target_node': {{'name': 'Dopamine D2 receptor'}}}},
        {{'source_node': {{'name': 'Dopamine D2 receptor'}}, 'relation': {{'name': 'gene_product_of'}}, 'target_node': {{'name': 'DRD2'}}}},
        {{'source_node': {{'name': 'Aniracetam'}}, 'relation': {{'name': 'increase_efficacy'}}, 'target_node': {{'name': 'Magnesium Sulfate'}}}},
        {{'source_node': {{'name': 'Magnesium Sulfate'}}, 'relation': {{'name': 'increase_efficacy'}}, 'target_node': {{'name': 'Paramethadione'}}}},
        {{'source_node': {{'name': 'Magnesium Sulfate'}}, 'relation': {{'name': 'increase_efficacy'}}, 'target_node': {{'name': 'Dihydrocodeine'}}}},
        {{'source_node': {{'name': 'Magnesium Sulfate'}}, 'relation': {{'name': 'increase_efficacy'}}, 'target_node': {{'name': 'Orvepitant'}}}}]"
    ]
}}

