# OpenReview forum: "Semantic-KG: Using Knowledge Graphs to Construct Benchmarks for Measuring Semantic Similarity"
_NeurIPS.cc/2025/Datasets_and_Benchmarks_Track — NeurIPS 2025 Datasets and Benchmarks Track poster_

### Official Review · Reviewer_aAHz · 2025-06-27

**Rating:** 4
**Confidence:** 4

**Summary:**

This paper proposes a novel method for generating semantic similarity benchmark datasets using knowledge graphs, which aims to address issues with existing benchmarks, such as high generation costs, lack of domain specificity, and ambiguous definitions of semantic equivalence. The proposed method generates pairs of semantically similar or dissimilar natural language statements by perturbing knowledge graphs, categorizing dissimilar pairs into four subtypes. The authors generated benchmark datasets across four domains (general knowledge, biomedicine, finance, biology) and conducted a comparative study of semantic similarity methods, including traditional NLP scores and LLM-as-a-judge predictions. The findings indicate that both the subtype of semantic variation and the domain of the benchmark impact the performance of semantic similarity methods, with no single method consistently outperforming others. The paper highlights the implications of using LLM-as-a-judge in detecting the semantic content of text.

**Additional Feedback:**

I would be positive to adjust my ratings based on the authors' rebuttals.

**Dataset Code Accessibility:**

Yes

**Ethical Considerations:**

No, there are no or only very minor ethics concerns

**Final Justification:**

After reviewing the author rebuttal and spending additional time thinking through the paper's contributions, I have decided to update my score.

**Resolved Issues:**

- The authors have added empirical results for NLI-based baselines (e.g., DeBERTa-MNLI), addressing my primary concern about incomplete comparisons. The reported results are appropriately interpreted, and the authors handled the application of NLI models carefully.
- They also provided quantitative and qualitative evaluations of the generated data's correctness and naturalness, including human annotation and textual analysis, which mitigates concerns about the reliability of the LLM-generated statements.
- The limitations of edge-level perturbations and the scope of real-world semantic variation are now better acknowledged in the revised limitations section, with discussion on how future work could expand the framework.

**Unresolved or Partially Addressed:**

- I still believe that structural perturbations on knowledge graphs, especially at the edge level, do not capture the full richness of real-world semantic drift. However, I acknowledge that this is a broader limitation of any automated benchmark and not specific to this paper.

**Weighting and Recommendation:**

While there are conceptual limitations that future work needs to address, I believe the proposed framework is technically solid, well-executed, and valuable for evaluating semantic similarity in a scalable and domain-adaptable way. The additional baselines and validation strengthen the paper. I therefore consider the contribution sufficient for acceptance and have adjusted my score accordingly.

**Limitations Weaknesses:**

1) The paper proposes four KG perturbation types (node removal, node replacement, edge removal, edge replacement) to simulate semantic variations. However, these perturbations primarily focus on the addition, deletion, or modification of factual knowledge. Real-world semantic differences are far more complex than these types, e.g., changes in tone, implied sarcasm, context-dependent meanings, or lack of common-sense reasoning, all of which are difficult to capture through simple KG structural perturbations. Therefore, this benchmark may not fully evaluate the capability of semantic similarity methods in handling these complex semantic phenomena. The paper does not provide sufficient argumentation to prove that these perturbation types are representative of all important semantic variations.

2) As the authors state in the limitations section partly, the paper extensively uses LLMs in both the statement generation and validation phases. While this increases automation, it also introduces the inherent limitations and potential biases of LLMs themselves. For example, the quality and diversity of LLM-generated statements might be limited by their training data and model capabilities, potentially leading to benchmark data that is not sufficiently "natural" or fails to cover the complexity of language. Furthermore, whether LLM's "reconstruction accuracy" in the validation phase can be fully equated to true semantic correctness remains debatable.

3) Most critically, the paper does not include any comparison against Natural Language Inference (NLI)-based models, despite their proven effectiveness in capturing semantic similarity in numerous recent works. In particular, models like `microsoft/deberta-base-mnli` and `microsoft/deberta-large-mnli`, fine-tuned on the MNLI dataset, have demonstrated robust performance in semantic understanding tasks, often surpassing traditional STS metrics in generalization and accuracy. While the authors briefly mention NLI in the related work section, they do not include any empirical evaluation involving these models. This omission weakens the validity of their comparative analysis and undermines the claimed superiority or generality of their benchmark. I strongly recommend the authors include experiments with such NLI-based models—both readily available on HuggingFace—as they represent a crucial baseline for any semantic similarity benchmark.

**Strengths Contributions:**

The primary contribution of this paper lies in proposing a knowledge graph-based framework for generating semantic similarity benchmark datasets. Its core advantage is the ability to generate statement pairs with controlled semantic variations through KG perturbations, thereby avoiding reliance on subjective human annotation. This partially addresses the issues of high generation costs and ambiguous definitions in existing semantic similarity benchmarks. The paper constructs and releases datasets in four diverse domains (general knowledge, biomedicine, finance, and biology), which hold some value for advancing cross-domain semantic similarity research. Furthermore, the paper provides a comparative analysis of LLM-as-a-judge and traditional NLP methods across different semantic variation types and domains, highlighting that LLMs do not consistently outperform traditional methods, offering useful insights for LLM applications in semantic similarity tasks.

---

> ### Author Rebuttal · Authors · 2025-07-30
>
> We would like to thank you for your thoughtful review. Your comments have provided key insights that have allowed us to make several important improvements to the quality of the manuscript.
>
> In the following paragraphs, we will address each of your comments in the order they were presented, detailing the changes we have made to the manuscript.
>
>
> ---
> ### **Reviewer Comment 1**
>
> > _[Real world semantics]_
>
> **Our Response:**
>
>
> _[Thank-you for raising this point, you are right to note that our benchmark will not cover the full scope of possible semantic variations. It is worth noting that even with our relatively simple semantic variations, the benchmark was still capable of identifying limitations of different semantic-similarity methods, highlighting that there is still utility to our approach. However we hope that by open-sourcing our benchmark, others in the community can build upon our framework and design new perturbation types that capture more complex semantic variations. We touched on this point in our limitations (lines 320-325) but have expanded on and provide the amended version below.]_
>
> [lines 320-325]
>
> > (2) Scope of Semantic Variations: Our current benchmark applies perturbations designed to create clear semantic distinctions. However real-world semantic variations are likely complex and not well captured by simple node or edge deletions and replacements, such as variations in intent or implicature, changes in tone, or context-dependent changes in meaning. Our framework is flexible to many perturbation types and future work might seek to apply more complex, realistic perturbations to generate benchmark data, however it is likely that some real-world semantic variations will not be well captured by structural knowledge-graph perturbations. Nonetheless our benchmark identified key limitations of different semantic-similarity methods highlighting that even simple semantic perturbations still have utility for evaluating these methods.
>
>
> ---
> ### **Reviewer Comment 2**
>
> > _[Limitations of LLM Generation]_
>
> **Our Response:**
>
>
> _[You are right to note that the use of an LLM for data generation has certain limitations. In your review you focus on the correctness and naturalness of the generated data. We will expand on each of these points individually. **Correctness**: As mentioned in your response, we make extensive use of a validation pipeline (Section 2.4) to ensure all generated responses are accurate representations of the underlying knowledge-graphs these are based on. Due to the high cardinality of the knowledge-graphs used in our dataset, it is very unlikely we could correctly reconstruct an exact match to the original subgraph from the generated response alone, unless this data was accurately contained within the textual statement. However, to further test for this, we ran a spot-check experiment and manually annotated a random sample of 100 generated statements (50 original statements, 50 perturbed statements) for correctness. We found that 99 out of the 100 were correct, and the only incorrect response represented the knowledge-graph correctly but included additional information about an edge that wasn’t present in the original subgraph. We therefore expect this component of the pipeline to be very robust, and don’t expect there to be significant correctness issues with our generated data. **Naturalness**: You are correct to question whether the generated statements sufficiently cover the complexity or style of real textual data. To test for this, we analysed our generated responses using several common NLP metrics related to readability, lexical characteristics, and syntactic complexity, and the results are provided below. Our generated responses match academic paper readability (Flesch 10-30 range) and exhibit high lexical diversity, typical of specialized academic writing. Our responses are similar in syntactic complexity to academic literature, though with shorter sentences and slightly lower noun ratios. It is worth noting that these metrics are not without their limitations, and “naturalness” is a subjective concept. To further test for this, two independent evaluators manually annotated 100 random samples (50 original statements, 50 perturbed statements) from our dataset for naturalness. A statement was considered “natural” if both annotators agreed. We found 75% of statements were considered “natural”, however there was little agreement between annotators (Cohen’s Kappa=0.04), indicating the challenges with assessing “naturalness”. Despite the limitations of both approaches, taken together we believe these provide some evidence that our generated data is similar in style to real textual data. Nonetheless, this remains a potential weakness with automated data generation approaches, and we have updated our limitations section to include this as a limitation, and have included our amended limitation below. We have also included an additional short section in the manuscript called Dataset Validation (section 3.1), where we describe the results of these annotation experiments, and include a brief description of the NLP metrics. We have included the results shown below in the Appendix.]_
>
> > Limitations on reliance on LLMs for generation: Our framework relies on LLMs to generate benchmark data. Though all generated samples are validated and grounded in a knowledge-graph, LLM generated textual data may still differ in distribution from real-world text, potentially introducing subtle biases. This framework is not intended to replace real human-labelled data, though still offers a valuable tool for identifying weaknesses in semantic-similarity methods, where data is scarce or human labelling too costly.
>
> ## Metrics Comparison
>
> ### Readability Metrics
>
> | Metric | Semantic-KG | Academic Papers | Technical Docs | Assessment |
> |--------|:----------:|:---------------:|:--------------:|:----------:|
> | **Flesch Reading Ease** | 16.2 ± 14.3 | 10-30 | 20-40 | Academic |
> | **Grade Level** | 15.0 ± 2.2 | 14-18 | 12-16 | Academic |
> | **Gunning Fog** | 19.4 ± 3.0 | 15-20 | 12-18 | Academic |
>
> ### Lexical Characteristics
>
> | Metric | Semantic-KG | Academic Papers | Technical Docs | Assessment |
> |--------|:----------:|:---------------:|:--------------:|:----------:|
> | **Type-Token Ratio** | 0.673 ± 0.110 | 0.50-0.70 | 0.45-0.65 | Academic |
> | **Lexical Density** | 0.693 ± 0.058 | 0.65-0.75 | 0.60-0.70 | Academic |
> | **Unique Words** | 48 ± 20 | 40-80 | 30-60 | Academic |
>
> ### Syntactic Complexity
>
> | Metric | Semantic-KG | Academic Papers | Technical Docs | Assessment |
> |--------|:----------:|:---------------:|:--------------:|:----------:|
> | **Avg Sentence Length** | 16.4 ± 3.9 words | 20-30 | 15-25 | Technical |
> | **Noun Ratio** | 0.232 ± 0.082 | 0.25-0.35 | 0.20-0.30 | Technical |
> | **Verb Ratio** | 0.095 ± 0.026 | 0.08-0.12 | 0.10-0.15 | Academic |
> | **Noun-Verb Ratio** | 2.44 | 2.5-4.0 | 2.0-3.0 | Technical |
> | **Content Word Density** | 38.0% | 35-45% | 40-50% | Academic |
>
> ---
> ### **Reviewer Comment 3**
>
> > _[NLI methods]_
>
> **Our Response:**
>
> _[Thank you for bringing this to our attention, this is an important point, and it was an oversight on our behalf due to time constraints, not to include these models in our results. We have now run a full evaluation using `microsoft/deberta-large-mnli` and include the results below. Please note, there are subtleties in how to apply this type of model to a semantic-similarity task. We found a bidirectional approach yielded the best results, whereby we tested entailment in both directions for each statement pair and only output a positive label when the output was “entailment” for both. These findings revealed some interesting performance characteristics. The model appeared to perform well for replacement-perturbations (though not consistently across all datasets) yet demonstrated weaker performance for deletion and removal perturbations. This is likely because replacement perturbations tend to introduce contradictions which these models are specifically optimised for. Thank you again for this valuable suggestion, and we will be sure to include these results in our camera-ready version alongside an additional baseline suggested by reviewer u3Nc.]_
>
> | dataset_name   | perturbation_type   |    f1 | 95% CI         |
> |:---------------|:--------------------|------:|:---------------|
> | oregano        | edge_replacement    | 0.928 | (0.874, 0.957) |
> | oregano        | edge_deletion       | 0.717 | (0.614, 0.793) |
> | oregano        | node_replacement    | 0.864 | (0.725, 0.940) |
> | oregano        | node_removal        | 0.715 | (0.596, 0.802) |
> | codex          | edge_replacement    | 0.700 | (0.515, 0.823) |
> | codex          | edge_deletion       | 0.675 | (0.515, 0.790) |
> | codex          | node_replacement    | 1.000 | (0.920, 1.000) |
> | codex          | node_removal        | 0.736 | (0.587, 0.840) |
> | findkg         | edge_replacement    | 0.954 | (0.817, 0.996) |
> | findkg         | edge_deletion       | 0.692 | (0.487, 0.836) |
> | findkg         | node_removal        | 0.779 | (0.645, 0.864) |
> | findkg         | node_replacement    | 0.954 | (0.817, 0.996) |
> | globi          | edge_replacement    | 0.692 | (0.559, 0.789) |
> | globi          | edge_deletion       | 0.672 | (0.548, 0.766) |
> | globi          | node_removal        | 0.714 | (0.604, 0.795) |
> | globi          | node_replacement    | 0.744 | (0.598, 0.840) |
>
> We would like to thank you again for your thoughtful review and suggestions which have contributed some noteworthy findings. We believe all your comments raise important points and limitations, and the resulting amendments have substantially improved the paper's quality. We hope that we have addressed your concerns, that the paper now meets your expectations for publication, and that you would kindly consider raising your score to indicate your suggestion for acceptance of this paper.

---

### Official Review · Reviewer_wMGc · 2025-06-28

**Rating:** 4
**Confidence:** 3

**Summary:**

This work presents a systematic way, utilizing KG, to generate a benchmark to semantic similarity benchmarks. It starts from a existing KG within a domain and extracts subgraphs from the KG to form individual data points. It generates positive data (semantically similar) by prompting LLM to generate different textual paraphrases of the same subgraph. It generates negative data (semantically dissimilar) by applying perturbations to the subgraphs and generating a textual description of the perturbed graphs. The scheme is applied to four distinct domain and it compares LLM-as-judge, Embedding models, and NLP approaches on the generated benchmark, highlighting non-transferrability of existing models across different domains and the surprising effectiveness of NLP approaches on certain perturbations.

**Dataset Code Accessibility:**

Yes

**Ethical Considerations:**

No, there are no or only very minor ethics concerns

**Final Justification:**

While the limitation still exists after discussion in the rebuttal, the authors present reasonable explanation, and the limitation does not significantly reduce the impact of the proposed method. I intend to keep my original rating.

**Limitations Weaknesses:**

- The replacement perturbation can be too simple. While the other four introduce subtle difference to the positive and negative pair, the replacement perturbation uses randomly sampled false node, and randomly sampled one can easily be uncorrelated to the original subgraph. Essentially, there might not be "hard negative" samples in node replacement, making this task not as challenging and useful, as can be seem in the higher node replacement performance.

- It appears that most subgraphs contain ~10 nodes and ~10 edges (some even smaller). One question is, why keep them small? Shorter responses could favor NLP approaches, which might not be ideal for evaluation for LLM-as-judge. I think this might be related to the inherent attribute of the original KG, but is also a potential limitation of the proposed approach.

**Strengths Contributions:**

The benchmark description and generation process is well-documented.

The proposed dataset and initial findings on the shortcomings of existing LLMs are surprising and critical to advance LLM-as-judge research, and consequently many LLM research field.

The proposed scheme is not limited to the generated four domains and can be potentially extend to any domains, especially with KG, and needs grounding on the relational data.

This is novel approach to incorporate KG information.

---

> ### Author Rebuttal · Authors · 2025-07-30
>
> We would like to express our sincere gratitude for your thoughtful comments on our paper and for acknowledging our systematic and novel approach to generating this benchmark. Your comments have highlighted some important areas of discussion.
>
> In the following paragraphs, we will address each of your comments in the order they were presented, detailing the changes we have made to the manuscript.
>
> ---
> ### **Reviewer Comment 1**
>
> > _[Replacement difficulty]_
>
> **Our Response:**
>
> _[Thank-you for raising this point about the limitations of node-replacement perturbations. In section 2.2 we describe how we mitigate the risk of producing a perturbed subgraph that is uncorrelated to the original subgraph, by only allowing node-replacements with nodes of the same type. We have amended this section to improve the clarity on this point:]_
>
> > Lines 118-121
> Node Replacement: A node is randomly replaced with another node and all edges to and from the old node are modified to point to and from the new node. To ensure the sampled node maintains logical consistency with the existing subgraph, the node is only replaced by nodes of same type. For example a node of type "Person" would only be replaced by another person.
>
> _[In future work we might explore methods to directly control the semantic closeness of candidate replacement nodes, using methods such as embedding models, to understand their impact on performance. We have amended our limitations and future work section to highlight this as a potential avenue for future research.]_
>
> > Future versions of this framework might explore techniques to control the semantic closeness of candidate replacement nodes, using methods such as embedding models, to understand how this impacts on the performance of different semantic-similarity methods.
>
> ---
> ### **Reviewer Comment 2**
>
> > _[Larger subgraphs]_
>
> **Our Response:**
>
> _[Many thanks for raising this issue. In fact when developing this framework we experimented with larger subgraphs, however we found the success rate during the reconstruction process was too low when the number of edges/triples was higher than around 12 nodes, and we simply couldn't generate a sufficiently large dataset in a reasonable time. We have included a plot in the Appendix demonstrating the relationship between subgraph size and reconstruction success, which shows a clear decline in success as the number of triples increases. This will be included in the camera-ready version as figures cannot be uploaded in our rebuttal. Although we didn't perform any systematic analysis, we found through manual inspection, that this failure appeared to result both from issues with generation: the LLMs were less likely to produce a statement that was a valid reflection of the underlying subgraph, and validation: the LLMs were unable to correctly reconstruct the subgraph, even when presented with a valid statement. As language-models improve, and as others develop on our framework and improve the validation process, we hope we can expand our datasets to include larger subgraphs. Despite this limitation, our benchmark was still capable of highlighting the limitations of all semantic-similarity methods for certain domains and/or perturbation types. We have added the following limitation to our manuscript to address this:]_
>
> > Limitation of subgraph-size: The success of the generation and validation pipeline in our framework declines at higher subgraph sizes (>12 triples), which limits the size of generated statements. As LLMs improve we hope to build on our framework to encompass larger subgraphs, which will allow for the generation of more complex semantic-similarity benchmarks.
>
> We would like to thank you again for your review. Your comments have highlighted some important limitations with our approach and by addressing these in our manuscript, we believe this has improved the quality of our paper. We hope that we have addressed your concerns, that the paper now meets your expectations for publication, and that you would kindly consider raising your score to indicate your suggestion for acceptance of this paper.

---

> > ### Comment · Reviewer_wMGc · 2025-08-05
> >
> > Thanks for acknowledging the limitations, and despite the limitations, the contribution is valuable and impactful.
> >
> > It is very intrigueing to see the recon-rate figure w.r.t. the graph size and the sudded decline for >12 triplets in future versions.

---

> > > ### Author Response · Authors · 2025-08-05
> > > **We Thank the Reviewer for Their Help Improving the Paper**
> > >
> > > We would like to express our sincere gratitude for your time spent reviewing, providing suggestions, and improving the paper. Your comments have helped the paper improve substantially. Thanks again for your time spent and all your help. Recon is indeed a research direction worthy of future work.

---

### Official Review · Reviewer_u3Nc · 2025-07-02

**Rating:** 5
**Confidence:** 3
**Ethics Flags:** Data privacy, copyright, and consent

**Summary:**

The paper introduces the **Semantic-KG framework**, a pipeline for generating semantically similar and dissimilar text pairs by applying targeted perturbations to subgraphs of knowledge graphs (KGs). The benchmark spans four domains: general knowledge, biomedicine, finance, and biology ([lines 12-19], [179-186]). The authors propose that by controlling semantic similarity through perturbations at the node and edge level ([lines 109-131]), they can produce fine-grained semantic variation without relying on costly human annotation ([lines 53-64]). This approach is used to benchmark both traditional NLP similarity metrics (e.g., ROUGE, BLEU) and LLM-as-a-judge approaches for measuring semantic similarity across domains ([lines 66-79]). The authors show that no single method performs best across all domains and perturbation types and that the type of semantic change (e.g., edge vs. node) significantly influences relative model performance.

**Additional Feedback:**

- **Incorrect Framing of Evaluation Methods**
Throughout the paper (e.g., [lines 236-237], Section 5.2, and Appendix D), metrics like ROUGE and BLEU are referred to as models, which is incorrect. These are string-overlap metrics, not semantic similarity models. This misclassification should be corrected throughout for technical accuracy. This can be fixed in the camera-ready version.

- **Figures need significant improvement**:
  - Figure 4 is visually cluttered and mixes metrics (e.g., ROUGE, BLEU) with models in the legend. These are not methods but evaluation metrics ([lines 236-237]).
  - Figure 2 (Semantic-KG framework) contains overly dense text, making it hard to follow at a glance.
  - Figures in Appendix D are poorly formatted and hard to interpret.

- Consider including SBERT, SimCSE, or Sentence-T5 as baselines in future iterations.

- Fix technical terminology around ROUGE/BLEU throughout the paper.

- Make sure all benchmarked models are cited properly and consistently.

- The submitted supplementary material is identical to the paper PDF. Please correct this to include additional content or assets.

**Dataset Code Accessibility:**

Yes

**Dataset Code Comments:**

The dataset can easily be loaded and the code works as well.

**Ethical Comments:**

Please ensure that all models benchmarked in the paper are properly cited.

**Ethical Considerations:**

Yes, there are ethics concerns that require attention by the authors

**Final Justification:**

The authors have addressed all the concerns that were raised. They have agreed to cite the models and implement the recommended changes to the figures. The authors have also included the missing baselines. They have revised the wording to avoid overstating the subjectivity of human judgments in semantic similarity benchmarks. Additionally, they have acknowledged the use of LLMs in the loop as a limitation. Although some limitations still persist, I believe the revisions have adequately addressed the concerns, and I am therefore raising my score to 5.

**Limitations Weaknesses:**

- **LLM-in-the-Loop Concerns**
  The authors use LLMs to generate and validate benchmark data and then evaluate other LLMs on this data, introducing a feedback loop that may reinforce model-specific biases or hallucinated errors rather than serve as a neutral benchmark ([lines 65-79]).

- **Missing citations of Benchmarked Models**
Several models included in experiments (e.g., Llama, Gemini) are not properly cited, contradicting the claim of completeness in the checklist.

- **Omission of Modern Baselines**
  Only classic metrics and embeddings are discussed ([lines 34-36]). There is no mention or benchmarking of state-of-the-art semantic similarity models like SBERT, SimCSE, Sentence-T5, which are widely used in STS tasks and should be strong baselines.

- **Subjectivity Claim Is Overstated**
The paper claims benchmarks like STS or MRPC rely on subjective human judgment without clear definitions ([lines 63–65]). However, STS datasets like SemEval 2017 use explicit scoring guidelines, e.g., 0-5 similarity scale with annotated examples. The paper should acknowledge these structured efforts and consider how such graded annotation schemes could be integrated into their current binary labeling framework.

**Strengths Contributions:**

- **Domain Coverage and Generalization Goal**
  The paper targets a significant gap in the semantic similarity evaluation space by introducing a benchmark spanning four domains: Codex (general knowledge), Oregano (biomedicine), FinDKG (finance), and Globi (biology) ([lines 179-186]).

- **Control Over Semantic Variation**
  By using graph perturbations (e.g., edge replacement, node deletion), the framework can systematically generate graded similarity and dissimilarity pairs, allowing for more fine-grained control over semantic content ([lines 109-131]). This bypasses the need for expensive human labeling ([lines 53-64]).

- **Reproducibility**
  The code and datasets are released and they work as described, and also appear to be well-structured. The benchmark is easy to load and extend ([lines 12-19]).

- **Empirical Insight on Model Behavior**
  The analysis in Section 5 highlights that LLMs outperform traditional methods in edge perturbations, while traditional metrics perform better in node-based perturbations ([lines 252-258]). This is a useful insight for real-world semantic similarity applications.

---

> ### Author Rebuttal · Authors · 2025-07-30
>
> Thank you for taking the time to review our paper, we appreciate the thoughtful and constructive feedback, and thank you for acknowledging the contribution of our method and empirical insights. We have revised the manuscript in light of your comments and believe the manuscript is stronger as a result of your feedback.
>
> In the following paragraphs, we will address each of your comments in the order they were presented, detailing the changes we have made to the manuscript.
>
> ---
> ### **Reviewer Comment 1**
>
> > _[LLM-in-the-loop]_
>
> **Our Response:**
>
> _[Many thanks for raising this point. It is true that there are certain limitations when using LLMs to generate data, though we introduce steps to mitigate some of these potential limitations. In your review you mention that LLMs might introduce hallucinated errors into our benchmark. We make extensive use of a validation pipeline (Section 2.4) to ensure all generated statements accurately reflect the content of the knowledge-graphs these statements are based upon. It is true that this validation pipeline uses LLMs in the loop, however it is highly unlikely an LLM could recover an exact match to the original subgraph using the generated response alone, unless this data was accurately represented in the text, due to the high cardinality of the knowledge-graphs. To further test this claim and validate our generated responses for correctness, we ran a spot-check experiment with 100 randomly selected response (50 original statements, 50 perturbed statements) and found that 99 out of the 100 were correct. The only incorrect response correctly represented the subgraph but included one piece of additional information about the edge, that wasn’t strictly present in the original subgraph. Another limitation of using LLMs, is that the generated content might differ in distribution from real-world textual data. In our response to reviewer aAHz, we describe additional analyses and experiments we performed to test the responses for lexical characteristics and “naturalness”. Finally, the flexibility of our framework allows for any LLM or generation method to be used both during generation and validation. In future iterations of our dataset, we might consider combining the responses from many different language-models, to mitigate potential biases arising from any individual model. We have included an additional short section (3.1 Dataset Validation), describing the results of our annotation experiments. Additionally, we have updated our limitations section to include this as a limitation.]_
>
> > Limitations on reliance on LLMs for generation: Our framework relies on LLMs to generate benchmark data. Though all generated samples are validated and grounded in a knowledge-graph, LLM generated textual data may still differ in distribution from real-world text, potentially introducing subtle biases. This framework is not intended to replace real human-labelled data, though still offers a valuable tool for identifying weaknesses in semantic-similarity methods, where data is scarce or human labelling too costly.
>
> ---
> ### **Reviewer Comment 2**
>
> > _[Missing citations]_
>
> **Our Response:**
>
> _[Many thanks for pointing out the missing citations. We have properly cited the Llama and Gemini models in our paper. The added citations will be visible in the camera ready version of the paper. We hope this resolves your ethics considerations as well.]_
>
> ---
> ### **Reviewer Comment 3**
>
> > _[Modern baselines]_
>
> **Our Response:**
>
> _[Thank-you for bringing these baselines to our attention. In light of your comments, we have now run a full evaluation on our benchmark, using Sentence-T5, and we attach the results below. As with many methods, we found performance to vary depending on perturbation-type and dataset, though on the whole it appeared that this method did not outperform the other embedding models. We will include these results in the final version to ensure our dataset is benchmarked against state-of-the-art methods. Additionally, we have included another baseline suggested by aAHz in our results. To the best of our knowledge, SBERT and SimCSE are architectures rather than specific models. We kindly ask the reviewer to clarify which specific models they believe would be valuable, and we can ensure we run experiments with these as well.]_
>
> | dataset_name   | perturbation_type   |    f1 | 95% CI         |
> |:---------------|:--------------------|------:|:---------------|
> | codex          | edge_replacement    | 0.667 | (0.401, 0.852) |
> | codex          | edge_deletion       | 0.800 | (0.593, 0.931) |
> | codex          | node_removal        | 0.971 | (0.849, 0.996) |
> | codex          | node_replacement    | 0.920 | (0.796, 0.980) |
> | oregano        | edge_replacement    | 0.810 | (0.728, 0.876) |
> | oregano        | edge_deletion       | 0.626 | (0.478, 0.758) |
> | oregano        | node_removal        | 0.955 | (0.845, 0.994) |
> | oregano        | node_replacement    | 0.973 | (0.861, 0.997) |
> | findkg         | edge_replacement    | 0.824 | (0.655, 0.931) |
> | findkg         | edge_deletion       | 0.571 | (0.340, 0.781) |
> | findkg         | node_removal        | 0.914 | (0.772, 0.976) |
> | findkg         | node_replacement    | 0.970 | (0.845, 0.996) |
> | globi          | edge_replacement    | 0.105 | (0.013, 0.315) |
> | globi          | edge_deletion       | 0.095 | (0.012, 0.289) |
> | globi          | node_removal        | 0.790 | (0.669, 0.877) |
> | globi          | node_replacement    | 0.714 | (0.559, 0.830) |
>
> ---
> ### **Reviewer Comment 4**
>
> > _[Subjectivity claim]_
>
> **Our Response:**
>
> _[Thank-you for raising this point. We agree that we overstated the degree to which human judgements are subjective in semantic-similarity benchmarks and have revised this statement as follows:]_
>
> > Lines 63-65
> > Several benchmark datasets exist to test a model's ability to detect semantic equivalence between text, such as the STS benchmark, Winograd, and MRPC \citep{cer2017semeval, levesque2012winograd, DBLP:journals/biodb/LiSJSWLDMWL16, dolan2005automatically}. There are also domain-specific benchmarks like BIOSSES for testing semantic equivalence in the biomedical domain \citep{souganciouglu2017biosses}. However, these benchmarks are often costly to generate, relying on expensive human judgement \citep{chiang2023can, chen2024humans}. Furthermore, it is often not clear how semantic equivalence is defined, with several of these benchmarks relying on subjective human judgement rather than a clear notion of semantic equivalence \citep{wu2024towards}. Despite efforts to reduce human judgment subjectivity via annotation guidelines and structured scores such as Likert scales \citep{cer2017semeval}, human annotations may still be conflicting or unavailable for a specific domain.
>
> Additionally we have revised our limitations to include suggestions for how graded annotation schemes might be incorporated into our benchmark.
>
> > Simple task setup: Our current benchmark only evaluates semantic-similarity methods in the simple binary setting, however graded annotation schemes inspired by human-labeled datasets \citep{cer2017semeval} might be incorporated into future versions of our dataset, using metrics such as perturbation-count or graph similarity measures \citep{6313167}.
>
> ---
> ### **Reviewer Comment 5
>
> > _[Incorrect framing]_
>
> **Our Response:**
>
> _[Thank you for raising this issue. We have updated the manuscript to ensure ROUGE and BLEU as labelled as metrics rather than models. These implemented changes will be visible in the camera ready version.]_
>
> ---
> ### **Reviewer Comment 6**
>
> > _[Figure improvement]_
>
> **Our Response:**
>
> _[Many thanks for describing the difficulties you encountered when you are reading the paper. We have updated the figures by implementing the following changes: (1) We have increased the proportion of figure size to legend size and label size to make it easier to read. (2) We have changed the color choice, which should make the bars more distinguishable. (3) We have reduced spacing between groups and made the bars larger. (4) We have supplemented the figure with a table such that the F scores can be perceived more clearly. (5) We have distinguished between the term "metric" and the term "model". (6) We have reformatted and relabelled the figures in Appendix D and Figure 2. All these changes will be reflected in the camera ready version of our paper as figures cannot be uploaded in our rebuttal.]_
>
> ---
> ### **Reviewer Comment 7**
>
> > _[Supp material]_
>
> **Our Response:**
>
> _[Thank you for bringing this to our attention. We do not have any additional supplementary materials beyond the Appendix of the main manuscript. This ensures we can cross-reference Appendix sections in the main body of the text. We will correct this in the camera-ready version to avoid any additional confusion.]_
>
> We would like to express our sincere gratitude once again for your review, as the quality of our paper is substantially improved thanks to your suggestions. We hope that these revisions based on your comments have addressed your concerns, that the paper now meets your expectations for publication, and that you would kindly consider raising your score to indicate your suggestion for acceptance of this paper.

---

> > ### Comment · Reviewer_u3Nc · 2025-08-02
> >
> > Thank you for your detailed responses to my concerns. I appreciate that the authors have incorporated the feedback positively. Most of my concerns have been addressed. The ethical issues have also been addressed, as the models have been cited. I agree that the LLM-in-the-loop should be addressed as a limitation. Regarding SBERT and SimCSE, the intent behind the comment was to suggest benchmarking against publicly available instantiations of these architectures (e.g., SimCSE-BERT-base-uncased), as they are strong baselines for semantic similarity tasks. I appreciate the inclusion of Sentence-T5 as a result of this suggestion and the invitation to clarify which variants would be useful. I believe this paper now meets the bar for acceptance. I am updating my score to a 5 and no longer have any major concerns with the content of the paper, as well as any ethical concerns.

---

> > > ### Author Response · Authors · 2025-08-05
> > > **We Thank the Reviewer for Their Help Improving the Paper**
> > >
> > > We would like to express our sincere gratitude for your time spent reviewing, providing suggestions, and improving the paper. Your comments have helped the paper improve substantially. Thanks again for your time spent and all your help.

---

### Official Review · Reviewer_bh1T · 2025-07-02

**Rating:** 5
**Confidence:** 4

**Summary:**

This paper proposes a new method for generating datasets for evaluation of semantic similarity judgments.
The basic idea is to create true statements from knowledge graphs, and create false statements by using perturbations on knowledge graphs.
The authors specify four types of perturbations: Node (entity) deletion or replacement, and Edge/Relation deletion or replacement.
An LLM is used to convert KG triples into natural language statements.
The resulting pairs are dichotomous – they are either equivalent (same meaning) or not.
While this is an weakness (evaluating gradations of meaning is not possible), it is also an advantage – it allows automated creation of datasets for semantic similarity of certain types.

**Dataset Code Accessibility:**

Yes

**Ethical Considerations:**

No, there are no or only very minor ethics concerns

**Final Justification:**

The authors have responded to my comments and the responses are sufficient. Wiht the promised improvement in the manusucript it is going to be a good and worthy paper.

**Limitations Weaknesses:**

1.
Figures 4 & 5.
The figures are too small and line/color distinctions are very difficult if not impossible to see.
And the confidence interval lines obscure things farther. It is simply not possible to see some of the results.
Consider improving the images – reduce the sizes of labels, make bars larger and reduce spacing between groups.
Consider providing tables with numbers – the differences in F scores cannot be perceived.

2.
It is not quite clear how many triples are selected for creating examples.
A standard approach with only triples  Node-Relation-Node would lead to very simple sentences.
However, examples like “Shakespeare was a playwright born in Stratford upon Avon” show that more than one triple is used for creating a sentence. This process needs to be clarified and explained a bit more.  Examples in the appendix (G.2) show that generations are quite complicated. For example “Both Norway and Colombia are members of...” (page 32, line 1179) shows that non-trivial logical relations are created during verbalization of KG data. This needs to be explained in the paper.


3.
The paper provides no statistical analyses of the results, it just shows some bar-charts (even in the appendices – just more bar-charts).  Some basic statistical analyses would be useful.


4.
Note that work with KGs relies on the ‘closed world assumption’ (https://en.wikipedia.org/wiki/Closed-world_assumption), meaning that things that are not in the KG are presumed false. However, this might sometimes lead to wrong results. It needs to be mentioned.

5.
Another point – the generation of false statements by perturbation on KGs is not entirely new thing. This kind of approach was used for many years for generating multiple-choice questions with distractors from KGs. For example
Tahani Alsubait, Bijan Parsia, and Ulrike Sattler (2016). Ontology-Based Multiple Choice Question Generation. Künstliche Intelligenz, 30, 183–188. doi:10.1007/s13218-015-0405-9
Vinu Ellampallil Venugopal, and P. Sreenivasa Kumar (2018). Automated Generation of Assessment Tests from Domain Ontologies. Semantic Web Journal, 8 (6), 1023-1047. https://www.semantic-web-journal.net/content/automated-generation-assessment-tests-domain-ontologies-1
Consider acknowledging this in related work.

**Strengths Contributions:**

The paper describes generation of datasets from four different KGs, coming from  different domains.

The paper then describes the performance of various LLMs, embedding systems and classic NLP methods for estimating semantic similarity (eventually dichotomous judgments).

The results indicate that performance varies by subject domain by type of perturbation. Overall, manipulation of edges reduces performance. The finding that LLMs are outperformed by traditional NLP is some cases is surprising.

---

> ### Author Rebuttal · Authors · 2025-07-30
>
> We would like to thank you for your thoughtful review. Your comments have been very helpful in improving the paper in several key areas.
>
> In the following paragraphs, we will address each of your comments in the order they were presented, detailing the changes we have made to the manuscript.
>
> ---
> ### **Reviewer Comment 1**
>
> > _[Figures difficult to read]_
>
> **Our Response:**
>
> _[Thank-you for raising issues you encountered when reading the paper. We have updated the figures by implementing the following changes: (1) We have increased the proportion of figure size to legend size and label size to make it easier to read. (2) We have changed the color choice, which should make the bars more distinguishable. (3) We have reduced spacing between groups and made the bars larger. (4) We have also supplemented the figure with a table such that the F scores can be perceived more clearly. All these changes will be reflected in the camera ready version of our paper as figures cannot be uploaded in our rebuttal.]_
>
> ---
> ### **Reviewer Comment 2**
>
> > _[Triple description]_
>
> **Our Response:**
>
> _[Many thanks for bringing this issue to our attention. The total number of triples in the subgraph is equal to number of edges, which is available in Tables 4 and 5 in Appendix C. We have added references to these tables in both 2.1 (Subgraph Sampling) and 2.3 (Response Generation) to make this more clear. Additionally, we have amended 2.3 (Response Generation) to make the process more clear. Our amendments are shown below:]_
>
> > Lines 107:
> > The total number of nodes and edges in the sampled subgraphs are shown in Table 4 and Table 4.
>
> > Lines 140:
> > In this stage we convert the subgraph and perturbed subgraphs into natural-language statements using an LLM. First, each subgraph is converted into a triple format, consisting of “source-node”, “relation”, and “target-node” tuples. The number of triples per subgraph is equal to the number of edges (see Tables 4 and 5). An LLM is then instructed to generate a natural-language statement using all provided triples. Few-shot examples are used to encourage the LLM to express logical relationships that may not be explicitly represented by individual triples. For example given the triples: ("Norway", "member of", "Organisation for the Prohibition of Chemical Weapons") and ("Colombia", "member of", "Organisation for the Prohibition of Chemical Weapons"), the LLM might choose to represent this as a single statement: "Both Norway and Colombia are members of the Organisation for the Prohibition of Chemical Weapons".
>
> ---
> ### **Reviewer Comment 3**
>
> > _[Statistical analyses]_
>
> **Our Response:**
>
> _[Thank-you for bringing our attention to the lack of statistical analysis. We included confidence-intervals in all results but have also added statistical testing for the camera-ready version. There are some nuances in how one should conduct these statistical analyses, due to the non-independence of different samples in the dataset (for example, those from the same dataset or the same perturbation-type). Our approach to statistical analysis here was to use a mixed-effects model with "model" as the fixed effect and "dataset" and "perturbation-type" as random-effects, to account for this non-independence. We ran this analysis on our data and have amended 5.2 to include the results from these analyses.]_
>
> > Stratification by perturbation-type revealed disparities in performance across all semantic similarity methods, depending on the type of perturbation applied with statistical analysis revealing a significant effect of perturbation type for both node-removal (z=2.294, p=0.022) and node-replacement (z=2.759, p=0.006). Many methods appear to underperform when distinguishing between statements that differ as a result of edge perturbations compared to node perturbations. Interestingly, the relative superiority of LLMs compared to classic NLP methods appears to depend on the perturbation-type. When perturbing edges, LLMs appear to match or out-perform traditional methods, especially state-of-the-art models, however for node perturbations, such as node-deletions, the traditional methods actually out-perform the majority of LLMs. Statistical analyses revealed a significant interaction effect between perturbation type and model for several LLMs such as GPT-4o (GPT4o x edge-replacement: z=2.405, p=0.016) and Gemini-1.5-pro (Gemini-1.5-pro x edge-replacement: z=2.027, p=0.043).  This has important implications for real-world applications of semantic-similarity methods. LLMs might be better suited for applications that require detecting variations in the relationships between entities in a statement, for example detecting contradictions in text. However traditional methods likely suffice for settings that simply involve detecting whether statements encompass the same entities.
>
> > {Stratification by dataset} also revealed disparities in performance, though these were less pronounced for LLMs than for perturbation-type disparities, and no significant effect on performance was observed for any dataset. All methods appeared to show strong performance on the general-knowledge dataset (Codex), with slightly lower performance on domain-specific datasets such as Globi and FinDKG. Performance disparities were higher among traditional NLP methods with Bertscore outperforming all methods including LLMs on Globi (Bertscore x Globi: z=2.08, p=0.38), yet a marked performance drop observed for the finance dataset (FinDKG) for methods such as ROUGE1 and ROUGEL, though this effect was not observed to be statistically significant (Rouge 1 x FindKG: z=-1.779, p=0.075). A closer inspection of results, (see Appendix \ref{app:findkg_extended_results}) reveals that this performance drop was largely driven by poor performance for edge-replacement perturbations, further highlighting the insufficiency of these methods to detect modifications in textual relationships. Additionally, the relative benefit of LLMs appeared to be domain-dependent with some LLMs under-performing for certain domains such as Globi and Oregano, whereas other LLMs such as Gemini-1.5 Pro, showed superior performance on domain-specific datasets like Oregano. (Gemini-1.5-pro x Oregano: z=0.154, p=0.045). These results highlight that semantic-similarity performance in one domain may not necessarily translate to performance in another. As more datasets are adopted into this framework, this will further elucidate the strengths and weaknesses of different methods within domain-specific applications.
>
>
> ### **Reviewer Comment 4**
>
> > _[Closed world assumption]_
>
> **Our Response:**
>
> _[You are right to raise this concern and this was an oversight on our behalf not to mention this as a key limitation within our framework. We've amended our Limitations section below and will add the following to our camera-ready version.]_
>
> > Limitations of Knowledge-Graphs for encoding semantic knowledge: One key limitation of using knowledge-graphs to encode semantic knowledge is that they inherently operate under a closed-world assumption \citep{Reiter1978} whereby missing information is assumed false, however this may not capture the incompleteness or evolving nature of the knowledge encoded in datasets generated using our framework. In future work, a more nuanced evaluation might explore how performance of different semantic-similarity methods varies under an open-world assumption.
>
> ---
> ### **Reviewer Comment 5**
>
> > _[Missing citations]_
>
> **Our Response:**
>
> _[Many thanks for pointing us to the related works! We have properly acknowledged and cited these works. The new citations will be visible in the camera ready version of the paper.]_
>
> We would like to express our sincere gratitude once again for your review. We hope that these revisions based on your comments have addressed your concerns, and that the paper now meets your expectations for publication.

---

> > ### Comment · Reviewer_bh1T · 2025-08-08
> >
> > I wanted to thank the authors for detailed responses to my concerns. The authors mention that the manuscript is/will be updated and the promised updates seem sufficient.

---

### Note · Authors · 2025-08-12

Dear Reviewers, AC, SAC,

We would like to thank you once again for the insightful reviews and feedback we have received during this review process. We appreciate the positive feedback we received for our work with reviewers recognising our significant contributions to the field of semantic-similarity evaluation, highlighting the following strengths:
* The novelty of our KG-based framework for generating semantic-similarity benchmark data (bh1T, u3NC, wMGc, aAHz)
* The cross-domain applicability of our framework (bh1T, u3NC, wMGc, aAHz)
* The ability of our framework to reveal performance variations across different models and perturbation types (bh1T, u3NC, wMGc, aAHz)
* The scalability and automation offered by our framework (u3NC, aAHz)

Your feedback has been invaluable in further strengthening our work with the following changes:

1. We have included additional baseline models: Sentence-T5 (a semantic-similarity model), and `microsoft/deberta-large-mnli` (an NLI model) (u3Nc, aAHz).
2. We performed a spot-check experiment, validating the robustness of our reconstruction pipeline (u3Nc, aAHz).
3. We conducted additional analyses and a spot-check experiment to assess the lexical characteristics and naturalness of our generated data. These results demonstrate the similarity of our generated data to real textual data (u3Nc, aAHz).
4. We included additional statistical analyses to bolster the claims presented in our results (Bh1T).
5. We conducted analysis to explore the relationship between reconstruction accuracy and subgraph size (wMGc).
6. We expanded the limitations section to more thoroughly address the limitations associated with automated text generation (bh1T, u3Nc, wMGc, aAHz).
7. We added an ethical disclosure section discussing the potential risks and ethical considerations associated with the use and misuse of our benchmark (TEc8).
8. We improved the presentation quality of the figures and tables, and the clarity of certain sections of the paper (bh1T, u3Nc).

We believe these revisions directly address the key concerns raises by the reviewers and have substantially improved the manuscript. We are pleased to note that reviewers u3Nc and aAHz have increased their scores to indicate their acceptance of the paper, and all reviewers now rate our paper above the acceptance threshold. All changes will be incorporated into the camera-ready version of our manuscript.

Thank you again for your time and valuable contributions.

Sincerely,

Authors

---

### Decision · Program_Chairs · 2025-09-18

**Decision:**

Accept (poster)

**Comment:**

This paper introduces Semantic-KG, a framework for systematically generating semantic similarity benchmarks by perturbing knowledge graphs and verbalizing subgraphs into text. The authors release datasets across four domains (general, biomedical, finance, biology) and evaluate a wide range of methods, including LLMs-as-judges, embedding models, and traditional metrics. The work highlights that no single method is consistently superior: LLMs excel at certain perturbations (edge-level), while embeddings and classic STS metrics perform better on others (node-level). Reviewers have raised some important concerns: (i) the reliance on LLMs for both generation and validation raises the possibility of model-induced artifacts, (ii) having only four perturbation types, missing richer linguistic and pragmatic phenomena, and (iii) Subgraphs capped at ~10 triples; limits benchmark complexity. The authors have adequately addressed the issues during their rebuttal. Overall, I feel that this paper provides a well-motivated, novel benchmark creation pipeline with broad domain coverage, a strong evaluation protocol, and thoughtful ethical considerations. While there are limitations (restricted perturbation scope, reliance on LLM validation), these are clearly acknowledged and framed as future work. The authors have been highly responsive in addressing reviewer concerns, significantly strengthening the submission. I recommend acceptance. This work will be of clear interest to both the evaluation/benchmarking community and practitioners seeking robust cross-domain similarity benchmarks. The authors should commit to include the main issues in their rebuttal into the main paper if the paper gets accepted.